# CiPR: An Efficient Framework with Cross-instance Positive Relations for Generalized Category Discovery

**Shaozhe Hao**                                                                 *szhao@cs.hku.hk*
*Department of Computer Science*
*The University of Hong Kong*

**Kai Han**[*]                                                                  *kaihanx@hku.hk*
*Department of Statistics & Actuarial Science*
*The University of Hong Kong*

**Kwan-Yee K. Wong**[*]                                                         *kykwong@cs.hku.hk*
*Department of Computer Science*
*The University of Hong Kong*

**Reviewed on OpenReview:** *https://openreview.net/forum?id=1fNcpcdr1o*

## Abstract

We tackle the issue of generalized category discovery (GCD). GCD considers the open-world problem of automatically clustering a partially labelled dataset, in which the unlabelled data may contain instances from both novel categories and labelled classes. In this paper, we address the GCD problem with an unknown category number for the unlabelled data. We propose a framework, named CiPR, to bootstrap the representation by exploiting Cross-instance Positive Relations in the partially labelled data for contrastive learning, which have been neglected in existing methods. To obtain reliable cross-instance relations to facilitate representation learning, we introduce a semi-supervised hierarchical clustering algorithm, named selective neighbor clustering (SNC), which can produce a clustering hierarchy directly from the connected components of a graph constructed from selective neighbors. We further present a method to estimate the unknown class number using SNC with a joint reference score that considers clustering indexes of both labelled and unlabelled data, and extend SNC to allow label assignment for the unlabelled instances with a given class number. We thoroughly evaluate our framework on public generic image recognition datasets and challenging fine-grained datasets, and establish a new state-of-the-art. Code: https://github.com/haoosz/CiPR

## 1 Introduction

By training on large-scale datasets with human annotations, existing machine learning models can achieve superb performance (*e.g.*, Krizhevsky et al. (2012)). However, the success of these models heavily relies on the assumption that they are only tasked to recognize images from the same set of classes with large-scale human annotations on which they are trained. This limits their application in the real open world where we will encounter data without annotations and from unseen categories. Indeed, more and more efforts have been devoted to dealing with more realistic settings. For example, semi-supervised learning (SSL) (Chapelle et al., 2006) aims at training a robust model using both labelled and unlabelled data from the same set of classes; few-shot learning (Snell et al., 2017) tries to learn models that can generalize to new classes with few annotated samples; open-set recognition (OSR) (Scheirer et al., 2012) learns to tell whether or not an unlabelled image belongs to one of the classes on which the model is trained. More recently, the problem of novel category discovery (NCD) (Han et al., 2019) has been introduced, which learns models to automatically partition unlabelled data from unseen categories by transferring knowledge from seen categories. One assumption in early NCD methods is that unlabelled images are all from unseen categories only. NCD

---

[*]Corresponding authors

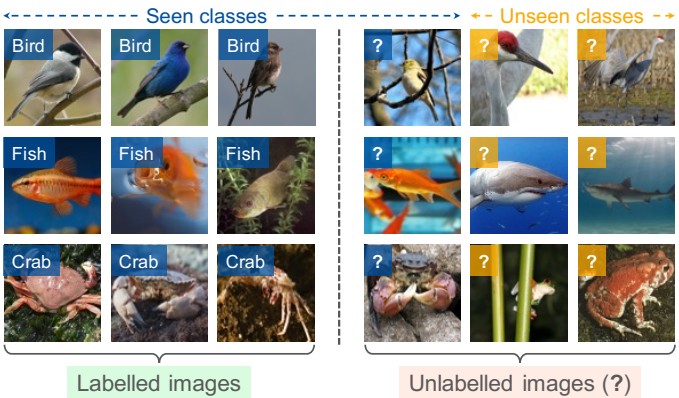

Figure 1: **Generalized category discovery**: given an image dataset with seen classes from a labelled subset, categorize the unlabelled images, which may come from seen and unseen classes.

has been recently extended to a more generalized setting, called generalized category discovery (GCD) (Vaze et al., 2022b), by relaxing the assumption to reflect the real world better, *i.e.*, unlabelled images are from both seen and unseen categories. An illustration of the GCD problem is shown in Fig. 1.

In this paper, we tackle the GCD problem by drawing inspiration from the baseline method (Vaze et al., 2022b). In Vaze et al. (2022b), a vision transformer model was first trained for representation learning using supervised contrastive learning on labelled data and self-supervised contrastive learning on both labelled and unlabelled data. With the learned representation, semi-supervised $k$-means (Han et al., 2019) was then adopted for label assignment across all instances. In addition, based on semi-supervised $k$-means, Vaze et al. (2022b) also introduced an algorithm to estimate the unknown category number for the unlabelled data by examining possible category numbers in a given range. However, this approach has several limitations. First, during representation learning, their method considers labelled and unlabelled data independently, and uses a stronger training signal for the labelled data which might compromise the representation of the unlabelled data. Second, their method requires a known category number for performing label assignment. Third, their category number estimation method is slow as it needs to run the clustering algorithm multiple times to test different category numbers.

To overcome the above limitations, we propose a new approach for GCD which does not require a known unseen category number and considers Cross-instance Positive Relations (CiPR) in unlabelled data for better representation learning. At the core of our approach is a novel semi-supervised hierarchical clustering algorithm, named selective neighbor clustering (SNC), that takes inspiration from the parameter-free hierarchical clustering method FINCH (Sarfraz et al., 2019). We introduce specific neighbor selection mechanisms for labeled and unlabeled examples. By carefully taking into account the properties of labeled and unlabeled images, our method generates higher quality pseudo labels which strengthen representation learning. SNC can not only generate reliable pseudo labels for cross-instance positive relations, but also estimate unseen category numbers without the need for repeated runs of the clustering algorithm. SNC builds a graph embedding all subtly selected neighbor relations constrained by the labelled instances, and produces clusters directly from the connected components of the graph. SNC iteratively constructs a hierarchy of partitions with different granularity while satisfying the constraints imposed by the labelled instances. With a one-to-one merging strategy, SNC can quickly estimate a reliable class number without repeated runs of the algorithm. This makes it significantly faster than Vaze et al. (2022b). Although there exist works that consider introducing pseudo positives in the literature for representation learning, such as Van Gansbeke et al. (2020); Dwibedi et al. (2021); Zhong et al. (2021), these methods only consider nearest neighbors whereas our method incorporates a novel semi-supervised clustering into training that fully exploits label information and produces more reliable pseudo labels with higher purity.

The main contributions of this paper can be summarized as follows: (1) we propose a new GCD framework, named CiPR, that exploits cross-instance positive relations in the partially labelled set to strengthen the connections among all instances, fostering the representation learning for better category discovery; (2) we introduce a semi-supervised hierarchical clustering algorithm, named SNC, that allows reliable pseudo label

generation during training and label assignment during testing; (3) we further leverage SNC for class number estimation by exploring intrinsic and extrinsic clustering quality based on a joint reference score considering both labelled and unlabelled data; (4) we comprehensively evaluate our framework on both generic image recognition datasets and challenging fine-grained datasets, and demonstrate state-of-the-art performance across the board.

## 2 Related work

Our work is related to novel/generalized category discovery, semi-supervised learning, and open-set recognition.

*Novel category discovery (NCD)* aims at discovering new classes in unlabelled data by leveraging knowledge learned from labelled data. It was formalized in Han et al. (2019) with a transfer clustering approach. Earlier works on cross-domain/task transfer learning (Hsu et al., 2018a;b) can also be adopted to tackle this problem. Han et al. (2020; 2021) proposed an efficient method called AutoNovel (aka RankStats) using ranking statistics. They first learned a good embedding using low-level self-supervised learning on all data followed by supervised learning on labelled data for higher-level features. They introduced robust ranking statistics to determine whether two unlabelled instances are from the same class for NCD. Several successive works based on RankStats were proposed. For example, Jia et al. (2021) proposed to use WTA hashing (Yagnik et al., 2011) for NCD in single- and multi-modal data; Zhao & Han (2021) introduced a NCD method with dual ranking statistics and knowledge distillation. Fini et al. (2021) proposed UNO which uses a unified cross entropy loss to train labelled and unlabelled data. Zhong et al. (2021) proposed NCL to retrieve and aggregate pseudo-positive pairs by exploring the nearest neighbors and generate hard negatives by mixing labelled and unlabelled samples in a contrastive learning framework. Chi et al. (2022) proposed meta discovery that links NCD to meta learning with limited labelled data. Joseph et al. (2022); Roy et al. (2022) consider NCD in the incremental learning scenarios. Vaze et al. (2022b) introduced generalized category discovery (GCD) which extends NCD by allowing unlabelled data from both old and new classes. They first finetuned a pretrained DINO ViT (Caron et al., 2021) with both supervised contrastive loss and self-supervised contrastive loss. Semi-supervised $k$-means was then adopted for label assignment. Cao et al. (2022); Rizve et al. (2022a;b) addressed the GCD problem from an open-world semi-supervised learning perspective. We draw inspiration from Vaze et al. (2022b) and develop a novel method to tackle GCD by exploring cross-instance relations on labelled and unlabelled data which have been neglected in the literature. Concurrent with our work, SimGCD (Wen et al., 2023) introduces a parametric classification approach for the GCD, while GPC (Zhao et al., 2023) proposes a method that jointly learns representations and estimates the class number for GCD.

*Semi-supervised learning (SSL)* aims at learning a good model by leveraging unlabelled data from the same set of classes as the labelled data (Chapelle et al., 2006). Various methods (Laine & Aila, 2017; Tarvainen & Valpola, 2017; Sohn et al., 2020; Zhang et al., 2021) have been proposed for SSL. The assumption of SSL that labelled and unlabelled data are from the same closed set of classes is often not valid in practice. In contrast, GCD relaxes this assumption and considers a more challenging scenario where unlabelled data can also come from unseen classes. Pseudo-labeling techniques are also widely adopted in SSL. Berthelot et al. (2019; 2020); Sohn et al. (2020); Rizve et al. (2020); Zhang et al. (2021) used one-hot pseudo labels generated from different augmentations. Rebalancing in pseudo labels was further studied in a class-imbalanced task (Wei et al., 2021) and a general problem (Wang et al., 2022). Tarvainen & Valpola (2017); Luo et al. (2018); Ke et al. (2019); Xie et al. (2020); Cai et al. (2021); Pham et al. (2021) applied teacher-student network to generate pseudo one-hot labels from the teacher. Our work is more related to methods that incorporate pseudo labels into contrastive learning (Chen et al., 2020b; Khosla et al., 2020), but they differ significantly from our method. A special case is SimCLR (Chen et al., 2020b;c), which utilizes self-supervised contrastive learning and relies on pseudo labels derived from augmented views. In contrast, our model further incorporates cross-instance positive relations. Unlike PAWS (Assran et al., 2021), we propose a new clustering method to assign pseudo labels instead of using a classifier based on supportive samples. In contrast to Li et al. (2021); Zheng et al. (2022); Bošnjak et al. (2022), our method does not require a memory bank or complex phases of distributional alignment and of unfolding and aggregation, and also enriches the positive samples compared to using the nearest neighbor like in SemPPL (Bošnjak et al., 2022).

*Open-set recognition (OSR)* aims at training a model using data from a known closed set of classes, and at test time determining whether or not a sample is from one of these known classes. It was first introduced

in Scheirer et al. (2012), followed by many works. For example, OpenMax (Bendale & Boult, 2016) is the first deep learning work to address the OSR problem based on Extreme Value Theory and fitting per-class Weibull distributions. RPL (Chen et al., 2020a) and ARPL (Chen et al., 2021) exploit reciprocal points for constructing extra-class space to reduce the risk of unknown. Recently, Vaze et al. (2022a) found the correlation between closed and open-set performance, and boosted the performance of OSR by improving closed-set accuracy. They also proposed Semantic Shift Benchmark (SSB) with a clear definition of semantic novelty for better OSR evaluation. Out-of-distribution (OOD) detection (Hendrycks & Gimpel, 2017; Liang et al., 2018; Hsu et al., 2020) is closely related to OSR but differs in that OOD detects general distribution shifts, while OSR focuses on semantic shifts. Several OOD methods have been proposed recently, such as Li et al. (2022) alleviating "posterior collapse" of hierarchical VAEs and Zheng et al. (2023) leveraging mistaken OOD generation for an auxiliary OOD task.

## 3 Methodology

### 3.1 Problem formulation

Generalized category discovery (GCD) aims at automatically categorizing unlabelled images in a collection of data in which part of the data is labelled and the rest is unlabelled. The unlabelled images may come from the labelled classes or new ones. This is a much more realistic open-world setting than the common closed-set classification where the labelled and unlabelled data are from the same set of classes. Let the data collection be $\mathcal{D} = \mathcal{D}_{\mathcal{L}} \cup \mathcal{D}_{\mathcal{U}}$, where $\mathcal{D}_{\mathcal{L}} = \{(x_i^{\ell}, y_i^{\ell})\}_{i=1}^{M} \in \mathcal{X} \times \mathcal{Y}_{\mathcal{L}}$ denotes the labelled subset and $\mathcal{D}_{\mathcal{U}} = \{(x_i^u, y_i^u)\}_{i=1}^{N} \in \mathcal{X} \times \mathcal{Y}_{\mathcal{U}}$ denotes the unlabelled subset with unknown $y_i^u \in \mathcal{Y}_{\mathcal{U}}$. Only a subset of classes contains labelled instances, $i.e.$, $\mathcal{Y}_{\mathcal{L}} \subset \mathcal{Y}_{\mathcal{U}}$. The number of labelled classes $N_{\mathcal{L}}$ can be directly deduced from the labelled data, while the number of unlabelled classes $N_{\mathcal{U}}$ is not known a priori.

To tackle this challenge, we propose a novel framework CiPR to jointly learn representations using contrastive learning by considering all possible interactions between labelled and unlabelled instances. Contrastive learning has been applied to learn representation in GCD, but without considering the connections between labelled and unlabelled instances (Vaze et al., 2022b) due to the lack of reliable pseudo labels. This limits the learned representation. In this paper, we propose an efficient semi-supervised hierarchical clustering algorithm, named selective neighbor clustering (SNC), to generate reliable pseudo labels to bridge labelled and unlabelled instances during training and bootstrap representation learning. With the generated pseudo labels, we can then train the model on both labelled and unlabelled data in a supervised manner considering all possible pairwise connections. We further extend SNC with a simple one-to-one merging process to allow cluster number estimation and label assignment on all unlabelled instances. An overview is shown in Fig. 2.

### 3.2 Joint contrastive representation learning

Contrastive learning has gained popularity as a technique for self-supervised representation learning (Chen et al., 2020b; He et al., 2020) and supervised representation learning (Khosla et al., 2020). These approaches rely on two types of instance relations to derive positive samples. Self-relations are obtained when the paired instances are augmented views from the *same image*, while cross-instance relations are obtained when the paired instances belong to the *same class*. For GCD, since data contains both labelled and unlabelled instances, a mix of self-supervised and supervised contrastive learning appears to be a natural fit, and good performance has been reported by Vaze et al. (2022b). However, cross-instance relations are only considered for pairs of labelled instances, but not for pairs of unlabelled instances and pairs of labelled and unlabelled instances. The learned representation is likely to be biased towards the labelled data due to the stronger learning signal provided by them. Meanwhile, the embedding spaces learned from cross-instance relations of labelled data and self-relations of unlabelled data might not be necessarily well aligned. These might explain why a much stronger performance on labelled data than on unlabelled data was reported in Vaze et al. (2022b). To mediate such a bias, we propose to introduce cross-instance relations for pairs of unlabelled instances and pairs of labelled and unlabelled instances in contrastive learning to bootstrap representation learning. We are the first to exploit labelled-unlabelled relations and unlabelled-unlabelled relations for unbiased representation learning in GCD. To this end, we propose an efficient semi-supervised hierarchical

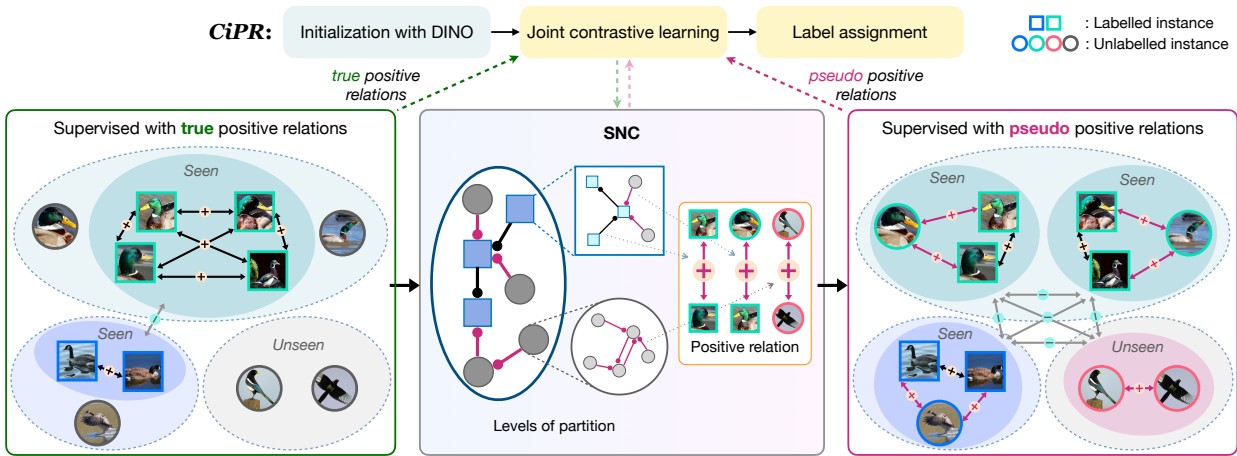

Figure 2: **Overview of our CiPR framework.** We first initialize ViT with pretrained DINO (Caron et al., 2021) to obtain a good representation space. We then finetune ViT by conducting joint contrastive learning with both true and pseudo positive relations in a supervised manner. True positive relations come from labelled data while pseudo positive relations of all data are generated by our proposed SNC algorithm. Specifically, SNC generates a hierarchical clustering structure. Pseudo positive relations are granted to all instances in the same cluster at one level of partition, further exploited in joint contrastive learning. With representations well learned, we estimate class number and assign labels to all unlabelled data using SNC with a one-to-one merging strategy.

clustering algorithm to generate reliable pseudo labels relating pairs of unlabelled instances and pairs of labelled and unlabelled instances, as detailed in Sec. 3.3. Next, we briefly review supervised contrastive learning (Khosla et al., 2020) that accommodates cross-instance relations, and describe how to extend it to unlabelled data.

**Contrastive learning objective.** Let $f$ and $\phi$ be a feature extractor and a MLP projection head. The supervised contrastive loss on labelled data can be formulated as

$$\mathcal{L}_i^s = -\frac{1}{|\mathcal{G}_{\mathcal{B}}(i)|} \sum_{q \in \mathcal{G}_{\mathcal{B}}(i)} \log \frac{\exp(\boldsymbol{z}_i^\ell \cdot \boldsymbol{z}_q^\ell / \tau_s)}{\sum_{n \in \mathcal{B}_\mathcal{L}, n \neq i} \exp(\boldsymbol{z}_i^\ell \cdot \boldsymbol{z}_n^\ell / \tau_s)} \tag{1}$$

where $\boldsymbol{z}^\ell = \phi(f(x^\ell))$, $\tau_s$ is the temperature, and $\mathcal{G}_{\mathcal{B}}(i)$ denotes other instances sharing the same label with the $i$-th labelled instance in $\mathcal{B}_\mathcal{L}$, which is the labelled subset in the mini-batch $\mathcal{B}$. Supervised contrastive loss leverages the true cross-instance positive relations between labelled instance pairs. To take into account the cross-instance positive relations for pairs of unlabelled instances and pairs of labelled and unlabelled instances, we extend the supervised contrastive loss on *all* data as

$$\mathcal{L}_i^a = -\frac{1}{|\mathcal{P}_{\mathcal{B}}(i)|} \sum_{q \in \mathcal{P}_{\mathcal{B}}(i)} \log \frac{\exp(\boldsymbol{z}_i \cdot \boldsymbol{z}_q / \tau_a)}{\sum_{n \in \mathcal{B}, n \neq i} \exp(\boldsymbol{z}_i \cdot \boldsymbol{z}_n / \tau_a)} \tag{2}$$

where $\tau_a$ is the temperature, $\mathcal{P}_{\mathcal{B}}(i)$ is the set of pseudo positive instances for the $i$-th instance in the mini-batch $\mathcal{B}$. The overall loss considering cross-instance relations for pairs of labelled instances, unlabelled instances, as well as labelled and unlabelled instances can then be written as

$$\mathcal{L} = \sum_{i \in \mathcal{B}} \mathcal{L}_i^a + \sum_{i \in \mathcal{B}_\mathcal{L}} \mathcal{L}_i^s \tag{3}$$

With the learned representation, we can discover classes with existing algorithms like semi-supervised $k$-means (Han et al., 2019; Vaze et al., 2022b). Besides, we further propose a new method in Sec. 3.4 based on our pseudo label generation approach as will be introduced next.

---

**Algorithm 1** Selective Neighbor Clustering (SNC)

---

1: **Preparation:**
2: Given labelled set $\mathcal{D}_\mathcal{L}$ and unlabelled set $\mathcal{D}_\mathcal{U}$, treat each instance in $\mathcal{D}_\mathcal{L} \cup \mathcal{D}_\mathcal{U}$ as a cluster $\mathbf{c}_i^0$ with the cluster centroid $\mu(\mathbf{c}_i^0)$ being each instance itself, forming the first partition $\Gamma^0 = \Gamma_\mathcal{L}^0 \cup \Gamma_\mathcal{U}^0$, where $\Gamma^0 = \{\mathbf{c}_i^0\}_{i=1}^{|\Gamma_\mathcal{L}^0|+|\Gamma_\mathcal{U}^0|}$.
3: **Main loop:**
4: $p \leftarrow 0$
5: **while** there are more than $N_\mathcal{L}$ clusters in $\Gamma^p$ **do**
6:     Initialize $\Gamma_\mathcal{L}^\star = \Gamma_\mathcal{L}^p$.
7:     **while** there exists $\kappa_i$ of $\mathbf{c}_i^p \in \Gamma_\mathcal{L}^p \cup \Gamma_\mathcal{U}^p$ not specified **do**
8:         **if** $\mathbf{c}_i^p \in \Gamma_\mathcal{L}^p$ **then**
9:             Initialize $\mathcal{Q} = \{\mathbf{c}_i^p\}$, $\Gamma_\mathcal{L}^\star = \Gamma_\mathcal{L}^\star \setminus \{\mathbf{c}_i^p\}$.
10:             **while** $|\mathcal{Q}| < \lambda$ **do**
11:                 $\kappa_i \leftarrow \arg\max_j \{\mu(\mathbf{c}_i^p) \cdot \mu(\mathbf{c}_j^p) \mid \mathbf{c}_j^p \in \Gamma_\mathcal{L}^\star, y_j^p = y_i^p\}$
12:                 $\Gamma_\mathcal{L}^\star \leftarrow \Gamma_\mathcal{L}^\star \setminus \{\mathbf{c}_{\kappa_i}^p\}$
13:                 $\mathcal{Q} \leftarrow \mathcal{Q} \cup \{\mathbf{c}_{\kappa_i}^p\}$
14:                 $\mathbf{c}_i^p \leftarrow \mathbf{c}_{\kappa_i}^p$
15:             **end while**
16:         **else**
17:             $\kappa_i \leftarrow \arg\max_j \{\mu(\mathbf{c}_i^p) \cdot \mu(\mathbf{c}_j^p) \mid \mathbf{c}_j^p \in \Gamma_\mathcal{L}^p \cup \Gamma_\mathcal{U}^p\}$
18:         **end if**
19:     **end while**
20:     Construct $A$ following Eq. (4) with selective neighbors, forming a new partition $\Gamma^{p+1} = \Gamma_\mathcal{L}^{p+1} \cup \Gamma_\mathcal{U}^{p+1}$.
21:     $p \leftarrow p + 1$
22: **end while**

---

### 3.3 Selective neighbor clustering

Although the concept of creating pseudo-labels may seem intuitive, effectively realizing it is a challenging task. Obtaining reliable pseudo-labels is a significant challenge in the GCD scenario, and ensuring their quality is of utmost importance. Naive approaches risk producing no performance improvement or even causing degradation. For example, an intuitive approach would be to apply an off-the-shelf clustering method like $k$-means or semi-supervised $k$-means to construct clusters and then obtain cross-instance relations based on the resulting cluster assignment. However, these clustering methods require a class number prior which is inaccessible in GCD task. Moreover, we empirically found that even with a given ground-truth class number, such a simple approach will produce many false positive pairs which severely hurt the representation learning. One way to tackle this problem is to overcluster the data to lower the false positive rate. FINCH (Sarfraz et al., 2019) has shown superior performance on unsupervised hierarchical overclustering, but it is non-trivial to extend it to cover both labelled and unlabelled data. Experiments show that FINCH will fail drastically if we simply include all the labelled data.

Inspired by FINCH, we propose an efficient semi-supervised hierarchical clustering algorithm, named SNC, with selective neighbor, which subtly makes use of the labelled instances during clustering.

**Preliminaries.** FINCH constructs an adjacency matrix $A$ for all possible pairs of instances $(i, j)$, given by

$$A(i, j) = \begin{cases} 1 & \text{if } j = \kappa_i \text{ or } \kappa_j = i \text{ or } \kappa_i = \kappa_j \\ 0 & \text{else} \end{cases}, \tag{4}$$

where $\kappa_i$ is the first neighbor of the $i$-th instance and is defined as

$$\kappa_i = \arg\max_j \{f(x_i) \cdot f(x_j) \mid x_j \in \mathcal{D}_\mathcal{U}\}, \tag{5}$$

where $f(\cdot)$ outputs an $\ell_2$-normalized feature vector and $\mathcal{D}_\mathcal{U}$ denotes an unlabelled dataset. A data partition can then be obtained by extracting connected components from $A$. Each connected component in $A$ corresponds to one cluster. By treating each cluster as a super instance and building the first neighbor adjacency matrix iteratively, the algorithm can produce hierarchical partitions.

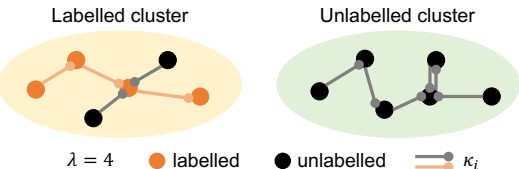

Figure 3: **Selective neighbor rules.** Left: the labelled instances constitute a chain (*rule 1*) with the length of $\lambda = 4$ (*rule 2*) and the nearest neighbors of unlabelled instances are labelled ones (*rule 3*). Right: the nearest neighbors of unlabelled instances are unlabelled ones (*rule 3*).

**Our approach.** First neighbor is designed for purely unlabelled data in FINCH. To make use of the labels in partially labelled data, a straightforward idea is to connect all labelled data from the same class by setting $A(i, j)$ to 1 for all pairs of instances $(i, j)$ from the same class. However, after filling $A(i, j)$ for pairs of unlabelled instances using Eq. (4), very often all instances become connected to a single cluster, making it impossible to properly partition the data. This problem is caused by having too many links among the labelled instances. To solve this problem, we would like to reduce the links between labelled instances while keeping labelled instances from the same class in the same connected component. A simple idea is to connect same labelled instances one by one to form a *chain*, which can significantly reduce the number of links. However, we found this still produces many incorrect links, resulting in low purity of the connected components. To this end, we introduce our selective neighbor for $\kappa_i$ to improve the purity of clusters while properly incorporating the labelled instances, constrained by the following rules. *Rule 1:* each labelled instance can only be the selective neighbor of another labelled instance once to ensure that labelled instances are connected in the form of chains; *Rule 2:* we limit the chain length to at most $\lambda$; *Rule 3:* the selective neighbor of an unlabelled instance depends on its actual distances to other instances, which can be either a labelled or an unlabelled instance. We illustrate how to cluster data with the selective neighbor rules in Fig. 3. Similar to FINCH, we can apply selective neighbor iteratively to produce hierarchical clustering results. We name our method SNC which is summarized in Algo. 1. The selective neighbor $\kappa_i$ of the $i$th sample $x_i$ can be simply formulated as

$$\kappa_i = \texttt{selective\_neigbor}(x_i, x_j \mid x_j \in \mathcal{D}_\mathcal{L} \cup \mathcal{D}_\mathcal{U}), \tag{6}$$

which corresponds to lines 7-19 in Algo. 1, in contrast to the first neighbor strategy in Eq. (5). For the chain length $\lambda$, we simply set it to the smallest integer greater than or equal to the square root of the number of labelled instances $n_\ell$ in each class, *i.e.*, $\lambda = \lceil \sqrt{n_\ell} \rceil$. $\lambda$ is automatically decided in each clustering hierarchy, motivated by a straight idea that it should be positively correlated with (but smaller than) the number of labelled instances $n_l$ at the current hierarchical level. The square root of $n_l$ is therefore a natural choice to balance the number of clustered instances in each cluster and the number of newly formed clusters. The chain length rule is applied to all classes with labelled instances, and at each hierarchy level. A proper chain length can therefore be dynamically determined based on the actual size of the labelled cluster and also the hierarchy level. We analyze different formulations of chain length in the Appx. A.

SNC produces a hierarchy of data partitions with different granularity. Except the bottom level, where each individual instance is treated as one cluster, every non-bottom level can be used to capture cross-instance relations for the level below, because each instance in the current level represents a cluster of instances in the level below. In principle, we can pick any non-bottom level to generate pseudo labels. To have a higher purity for each cluster, it is beneficial to choose a relatively low level that overclusters the data. Hence, we choose a level that has a cluster number notably larger than the labelled class number (*e.g.*, 2× more). Meanwhile, the level should not be too low as this will provide much fewer useful pairwise correlations. In our experiment, we simply pick the third level from the bottom of the hierarchy, which consistently shows good performance on all datasets. We discuss the impact of the picked level in the Appx. A. During the training stage, we enhance the joint representation learning by updating the pseudo labels using the data partition from the third level at the beginning of each training epoch. This iterative process helps to improve the clustering quality and allows for self-evolution of the pseudo labels.

### 3.4 Class number estimation and label assignment

Once a good representation is learned, we can then estimate the class number and determine the class label assignment for all unlabelled instances.

---

**Algorithm 2** One-to-one merging

---

1: **Preparation:**
2: Get initial partitions $S = \{\Gamma^p\}_{p=0}$ by SNC and a cluster number range $[N_e, N_o]$. Note that the merging is from $N_o$ to $N_e$ and $N_o > N_e$.
3: **Partition initialization:**
4: Find $\Gamma^t \in S$ satisfying $|\Gamma^t| > N_o$ and $|\Gamma^{t+1}| \leq N_o$.
5: **Merging:**
6: **while** $|\Gamma^t| > N_e$ **do**
7: $\quad (i, j) \leftarrow \arg\min_{i,j} \{\mu(\mathbf{c}_i^t) \cdot \mu(\mathbf{c}_j^t) \mid \mathbf{c}_i^t, \mathbf{c}_j^t \in \Gamma^t, y_i^t = y_j^t \text{ if } \mathbf{c}_i^t, \mathbf{c}_j^t \in \Gamma_{\mathcal{L}}^t\}$
8: $\quad$ Merge $\mathbf{c}_i^t$ and $\mathbf{c}_j^t$, forming a new partition $\Gamma^\star$.
9: $\quad$ Update current partition $\Gamma^t \leftarrow \Gamma^\star$.
10: **end while**
11: **Output**:
12: Obtain a specific partition $\Gamma^t$ of $N_e$ clusters, *i.e.*, $|\Gamma^t| = N_e$.

---

**Class number estimation.** When the class number is unknown, existing methods based on semi-supervised $k$-means need to first estimate the unknown cluster number before they can produce the label assignment. To estimate the unknown cluster number, Han et al. (2019) proposed to run semi-supervised $k$-means on all the data while dropping part of the labels for clustering performance validation. Though effective, this algorithm is computationally expensive as it needs to run semi-supervised $k$-means on all possible cluster numbers. Vaze et al. (2022b) proposed an improved method with Brent's optimization (Brent, 1971), which increases the efficiency. In contrast, SNC is a hierarchical clustering method that can automatically produce hierarchical cluster assignments at different levels of granularity. Therefore, it does not require a given class number in clustering. For practical use, one can pick any level of assignment based on the required granularity. To identify a reliable level of granularity, we conduct class number estimation by applying SNC with a joint reference score considering both labelled and unlabelled data. Specifically, we split the labelled data $\mathcal{D}_{\mathcal{L}}$ into two parts $\mathcal{D}_{\mathcal{L}}^l$ and $\mathcal{D}_{\mathcal{L}}^v$. We run SNC on the full dataset $\mathcal{D}$ treating $\mathcal{D}_{\mathcal{L}}^l$ as labelled and $\mathcal{D}_{\mathcal{U}} \cup \mathcal{D}_{\mathcal{L}}^v$ as unlabelled. We jointly measure the unsupervised intrinsic clustering index (such as silhouette score (Rousseeuw, 1987)) on $\mathcal{D}_{\mathcal{U}}$ and the extrinsic clustering accuracy on $\mathcal{D}_{\mathcal{L}}^v$. We obtain a joint reference score $s_c$ by simply multiplying them together after min-max scaling to achieve the best overall measurement on the labelled and unlabelled subsets. We choose the level in SNC hierarchy with the maximum $s_c$. The cluster number in the chosen level can be regarded as the estimated class number. To achieve more accurate class number estimation, we further introduce a simple *one-to-one merging strategy*. Namely, from the level below the chosen one to the level above the chosen one, we merge the clusters successively. At each merging step, we simply merge the two closest clusters. We identify the merge that gives the best reference score $s_c$ and its cluster number is considered as our estimated class number which we denote as $K_{1t1}$. SNC with the one-to-one merging strategy can carry out class number estimation with one single run of hierarchical clustering, which is significantly more efficient than the methods based on semi-supervised $k$-means (Han et al., 2019; Vaze et al., 2022a).

**Label assignment.** With a given (estimated) class number, we can obtain the label assignment by adopting semi-supervised $k$-means like Han et al. (2019); Vaze et al. (2022b) or directly using our proposed SNC. Since SNC is a hierarchical clustering algorithm and the cluster number in each hierarchy level is determined automatically by the intrinsic correlations of the instances, it might not produce a level of partition with the exact same cluster number as the known class number. To reach the estimated class number $K_{1t1}$, we therefore reuse the one-to-one merging strategy. Specifically, we begin by executing SNC to produce hierarchical partitions and identify the partition level that contains the closest cluster count greater than the given class number. We then employ one-to-one merging to successively merge the two closest clusters at each iteration until the target number $K_{1t1}$ is reached. Note that during the one-to-one merging process, clusters belonging to different labeled classes are not allowed to merge. The predicted cluster indices can be therefore retrieved from the final partition. The merging process is summarized in Algo. 2. Lastly, following Vaze et al. (2022b), we use the Hungarian algorithm (Kuhn, 1955) to find an optimal linear assignment from the predicted cluster indices to the ground-truth labels, which gives the final label assignment.

Table 1: **Results on generic image recognition datasets.** Our results are averaged over 5 runs.

| Classes | CIFAR-10 | | | CIFAR-100 | | | ImageNet-100 | | |
|---|---|---|---|---|---|---|---|---|---|
| | All | Seen | Unseen | All | Seen | Unseen | All | Seen | Unseen |
| RankStats+ (Han et al., 2021) | 46.8 | 19.2 | 60.5 | 58.2 | 77.6 | 19.3 | 37.1 | 61.6 | 24.8 |
| UNO+ (Fini et al., 2021) | 68.6 | **98.3** | 53.8 | 69.5 | 80.6 | 47.2 | 70.3 | **95.0** | 57.9 |
| ORCA (Cao et al., 2022) | 97.3 | 97.3 | 97.4 | 66.4 | 70.2 | 58.7 | 73.5 | 92.6 | 63.9 |
| Vaze et al. (2022b) | 91.5 | 97.9 | 88.2 | 70.8 | 77.6 | 57.0 | 74.1 | 89.8 | 66.3 |
| Ours (CiPR) | **97.7** | 97.5 | **97.7** | **81.5** | **82.4** | **79.7** | **80.5** | 84.9 | **78.3** |

Table 2: **Results on fine-grained image recognition datasets.** Our results are averaged over 5 runs.

| Classes | CUB-200 | | | SCars | | | Herbarium19 | | |
|---|---|---|---|---|---|---|---|---|---|
| | All | Seen | Unseen | All | Seen | Unseen | All | Seen | Unseen |
| RankStats+ (Han et al., 2021) | 33.3 | 51.6 | 24.2 | 28.3 | 61.8 | 12.1 | 27.9 | **55.8** | 12.8 |
| UNO+ (Fini et al., 2021) | 35.1 | 49.0 | 28.1 | 35.5 | **70.5** | 18.6 | 28.3 | 53.7 | 14.7 |
| ORCA (Cao et al., 2022) | 35.0 | 35.6 | 34.8 | 32.6 | 47.0 | 25.7 | 24.6 | 26.5 | 23.7 |
| Vaze et al. (2022b) | 51.3 | 56.6 | 48.7 | 39.0 | 57.6 | 29.9 | 35.4 | 51.0 | 27.0 |
| Ours (CiPR) | **57.1** | **58.7** | **55.6** | **47.0** | 61.5 | **40.1** | **36.8** | 45.4 | **32.6** |

## 4 Experiments

### 4.1 Experimental setup

**Data and evaluation metric.** We evaluate our method on three generic image classification datasets, namely CIFAR-10 (Krizhevsky et al., 2009), CIFAR-100 (Krizhevsky et al., 2009), and ImageNet-100 (Deng et al., 2009). ImageNet-100 refers to randomly subsampling 100 classes from the ImageNet dataset. We further evaluate on three more challenging fine-grained image classification datasets, namely Semantic Shift Benchmark (Vaze et al., 2022a) (SSB includes CUB-200 (Wah et al., 2011) and Stanford Cars (Krause et al., 2013)) and long-tailed Herbarium19 (Tan et al., 2019). We follow Vaze et al. (2022b) to split the original training set of each dataset into labelled and unlabelled parts. We sample a subset of half the classes as seen categories. 50% of instances of each labelled class are drawn to form the labelled set, and all the rest data constitute the unlabeled set. The model takes all images as input and predicts a label assignment for each unlabelled instance. For evaluation, we measure the clustering accuracy by comparing the predicted label assignment with the ground truth, following the protocol of Vaze et al. (2022b).

**Implementation details.** We follow Vaze et al. (2022b) to use the ViT-B-16 initialized with pretrained DINO (Caron et al., 2021) as our backbone. The output `[CLS]` token is used as the feature representation. Following the standard practice, we project the representations with a non-linear projection head and use the projected embeddings for contrastive learning. We set the dimension of projected embeddings to 65,536 following Caron et al. (2021). At training time, we feed two views with random augmentations to the model. We only fine-tune the last block of the vision transformer with an initial learning rate of 0.01 and the head is trained with an initial learning rate of 0.1. We update the pseudo labels with SNC at the beginning of each training epoch. All methods are trained for 200 epochs with a cosine annealing schedule. For our method, the temperatures of two supervised contrastive losses $\tau_s$ and $\tau_a$ are set to 0.07 and 0.1 respectively. For class number estimation, we set $|\mathcal{D}_{\mathcal{L}}^l|:|\mathcal{D}_{\mathcal{L}}^v| = 6{:}4$ for CIFAR-10 and $|\mathcal{D}_{\mathcal{L}}^l|:|\mathcal{D}_{\mathcal{L}}^v| = 8{:}2$ for the other datasets. The reason we set $|\mathcal{D}_{\mathcal{L}}^l|:|\mathcal{D}_{\mathcal{L}}^v| = 6{:}4$ is that CIFAR-10 has 5 labeled classes. If we were to set the ratio to 8:2, there would be only 1 class remaining in $\mathcal{D}_{\mathcal{L}}^v$, which would diminish its effectiveness in providing a reference clustering accuracy for validation. All experiments are conducted on a single RTX 3090 GPU.

### 4.2 Comparison with the state-of-the-art

We compare CiPR with four strong GCD baselines: *RankStats+* and *UNO+*, which are adapted from RankStats (Han et al., 2021) and UNO (Fini et al., 2021) that are originally developed for NCD, the state-of-the-art GCD method of Vaze et al. (2022b), and *ORCA* (Cao et al., 2022) which addresses GCD

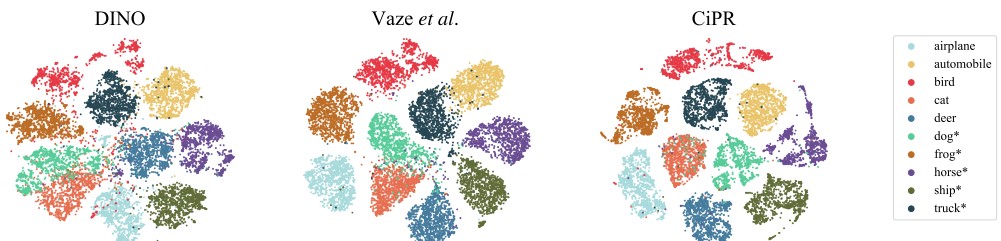

Figure 4: **Visualization on CIFAR-10.** We conduct t-SNE projection on features extracted by raw DINO, GCD method of Vaze et al. (2022b) and our CiPR. We randomly sample 1000 images of each class from CIFAR-10 to visualize. Unseen categories are marked with *.

Table 3: **Estimation of class number in unlabelled data.**

|  | Method | CIFAR-10 | CIFAR-100 | ImageNet-100 | CUB-200 | SCars | Herbarium19 |
|---|---|---|---|---|---|---|---|
| Ground truth | — | 10 | 100 | 100 | 200 | 196 | 683 |
| Estimate (error) | Vaze et al. (2022b) | 9 (10%) | 100 (0%) | 109 (9%) | 231 (16%) | 230 (17%) | 520 (24%) |
|  | Ours (CiPR) | 12 (20%) | 103 (3%) | 100 (0%) | 155 (23%) | 182 (7%) | 490 (28%) |
| Runtime | Vaze et al. (2022b) | 15394s | 27755s | 64524s | 7197s | 8863s | 63901s |
|  | Ours (CiPR) | 102s | 528s | 444s | 126s | 168s | 1654s |

from a semi-supervised learning perspective. As ORCA uses a different backbone model and data splits, for fair comparison, we retrain ORCA with ViT model using the official code on the same splits.

In Tab. 1, we compare CiPR with others on the generic image recognition datasets. CiPR consistently outperforms all others by a significant margin. For example, CiPR outperforms the state-of-the-art GCD method of Vaze et al. (2022b) by 6.2% on CIFAR-10, 10.7% on CIFAR-100, and 6.4% on ImageNet-100 for 'All' classes, and by 9.5% on CIFAR-10, 22.7% on CIFAR-100, and 12.0% on ImageNet-100 for 'Unseen' classes. This demonstrates cross-instance positive relations obtained by SNC are effective to learn better representations for unlabelled data. Due to the fact that a linear classifier is trained on 'Seen' classes, UNO+ shows a strong performance on 'Seen' classes, but its performance on 'Unseen' ones is significantly worse. In contrast, CiPR achieves comparably good performance on both 'Seen' and 'Unseen' classes, without biasing to the labelled data.

In Tab. 2, we further compare our method with others on fine-grained image recognition datasets, in which the difference between different classes are subtle, making it more challenging for GCD. Again, CiPR consistently outperforms all other methods for 'All' and 'Unseen' classes. On CUB-200 and SCars, CiPR achieves 5.8% and 8.0% improvement over the state-of-the-art for 'All' classes. For the challenging Herbarium19 dataset, which contains many more classes than other datasets and has the extra challenge of long-tailed distribution, CiPR still achieves an improvement of 1.4% and 5.6% for 'All' and 'Unseen' classes. Both RankStats+ and UNO+ show a strong bias to the 'Seen' classes.

In Fig. 4, we visualize the t-SNE projection of features extracted by DINO (Caron et al., 2021), GCD method of Vaze et al. (2022b), and our method CiPR, on CIFAR-10. Both Vaze et al. (2022b) and our features are more discriminative than DINO features. The method of Vaze et al. (2022b) captures better representations with more separable clusters, but some seen categories are confounded with unseen categories, *e.g.*, cat with dog and automobile with truck, while CiPR features show better cluster boundaries for seen and unseen categories, further validating the quality of our learned representation.

### 4.3 Estimating the unknown class number

In Tab. 3, we report our estimated class numbers on both generic and fine-grained datasets using the joint reference score $s_c$ as described in Sec. 3.4. Overall, CiPR achieves comparable results with the method of Vaze et al. (2022b), but it is far more efficient (40-150 times faster) and also does not require a list of predefined possible numbers. Even for the most difficult Herbarium19 dataset, CiPR only takes a few minutes to finish, while it takes more than an hour for a single run of $k$-means due to the large class number, let alone multiple

Table 4: **Label assignment results under different estimated class numbers.** We use the same protocol to evaluate Vaze et al. (2022b) and our method on the estimated class numbers by different methods. Here, $K_{km}$ and $K_{1t1}$ denote using the estimated class numbers by Vaze et al. (2022b) and ours respectively.

| Estimated #class | Method | CIFAR-10 | | | CIFAR-100 | | | ImageNet-100 | | | CUB-200 | | | SCars | | | Herbarium19 | | |
|---|---|---|---|---|---|---|---|---|---|---|---|---|---|---|---|---|---|---|---|
| | | All | Seen | Unseen | All | Seen | Unseen | All | Seen | Unseen | All | Seen | Unseen | All | Seen | Unseen | All | Seen | Unseen |
| $K_{km}$ | Vaze et al. (2022b) | 79.4 | **98.9** | 69.6 | 70.8 | 77.6 | 57.0 | 71.7 | 69.8 | 72.7 | 53.5 | **64.6** | 48.0 | 41.9 | **68.2** | 29.3 | 15.2 | 19.8 | 13.0 |
| | Ours (CiPR) | **85.0** | 97.3 | **78.8** | **81.5** | **82.4** | **79.7** | **78.6** | **81.2** | **77.3** | **55.7** | 56.8 | **55.2** | **45.3** | 58.9 | **38.7** | **36.8** | **43.8** | **33.3** |
| $K_{1t1}$ | Vaze et al. (2022b) | 83.6 | **98.2** | 76.3 | 75.1 | **82.4** | 60.3 | 74.1 | **89.8** | 66.3 | 48.3 | 55.8 | 44.6 | 40.5 | **64.8** | 28.8 | 15.5 | 19.9 | 13.4 |
| | Ours (CiPR) | **90.3** | 97.3 | **86.8** | **80.9** | 82.1 | **78.4** | **80.5** | 84.9 | **78.3** | **53.1** | **56.8** | **51.2** | **42.4** | 55.4 | **36.2** | **37.3** | **43.6** | **34.2** |

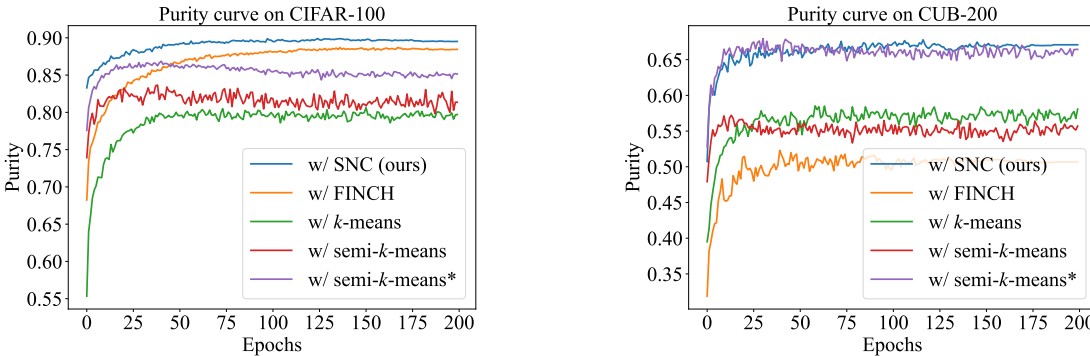

Figure 5: **Purity curve.** We plot the purity of all compared clustering methods throughout training.

runs from a predefined list of possible class numbers. Both methods have similar memory costs, as reported in the Appx. D. We also present the evaluation results based on the different settings of the estimated class numbers in Tab. 4. Our method consistently outperforms Vaze et al. (2022b) on all datasets, demonstrating its superior performance.

## 4.4 Ablation study

**Approaches to generating positive relations.** In Tab. 5, we compare our SNC with multiple different approaches to generating positive relations for joint contrastive learning, including directly using nearest neighbor in every mini-batch and conducting various clustering algorithms to obtain pseudo labels, *e.g.*, FINCH (Sarfraz et al., 2019), $k$-means (MacQueen et al., 1967), and semi-supervised $k$-means (Han et al., 2019; Vaze et al., 2022b). Non-hierarchical clustering methods ($k$-means and semi-supervised $k$-means) require a given cluster number. For $k$-means, we use the ground-truth class number. For semi-supervised $k$-means, we use both the ground truth and the overclustering number (twice the ground truth). For generating pseudo positive relations, our method achieves best performance among all approaches. FINCH performs great on CIFAR-100 but degrades on CUB-200. We hypothesize that because FINCH is purely unsupervised without leveraging labelled data, it fails to generate reliable pseudo labels of more semantically similar instances on fine-grained CUB-200. Overclustering semi-supervised $k$-means achieves comparable performance on CUB-200 but performs bad on CIFAR-100. This might be caused by intrinsic poorer performance of semi-supervised $k$-means compared to proposed SNC, which results in worse pseudo labels. We further report the mean purity curve of pseudo labels generated by all clustering methods throughout training process in Fig. 5. We can observe that pseudo labels produced by SNC remain the highest purity on both datasets throughout the entire training process.

We evaluate performance using both SNC and semi-supervised $k$-means for comparison. SNC reaches higher accuracy than semi-supervised $k$-means at test time. This advantage stems from the hierarchical design of SNC. In semi-supervised $k$-means, the number of labeled centroids remains fixed throughout all iterations and it only considers distances between instances and centroids. This may prevent certain correct instance grouping sometimes. For instance, if an unlabeled instance is close to a labeled instance but far from the centroid of the labeled cluster, it may not be assigned to that class. This limitation can have a negative impact on the overall quality of the clustering results. On the other hand, SNC is a hierarchical method that takes into account local distribution structures at each hierarchical layer for both labeled and unlabeled data. In the scenario mentioned above, where an unlabeled instance is close to a labeled instance but distant from

Table 5: **Results using different approaches to generating positive relations.** Semi-$k$-means$^\star$ denotes using semi-supervised $k$-means with an overclustering class number ($2 \times$ ground truth). We test with two clustering methods: SNC (normal font, left) and semi-supervised $k$-means (smaller font, right).

| | CIFAR-100 | | | CUB-200 | | |
|---|---|---|---|---|---|---|
| Classes | All | Seen | Unseen | All | Seen | Unseen |
| w/ nearest neighbor | 80.4 74.9 | **82.9** 77.5 | 75.5 69.7 | 51.9 45.8 | 56.7 48.2 | 49.5 44.6 |
| w/ FINCH | 81.4 76.6 | 81.7 75.7 | **80.7** 78.6 | 51.4 47.9 | 51.8 45.1 | 51.3 49.3 |
| w/ k-means | 76.7 72.2 | 77.1 70.4 | 75.7 75.8 | 52.8 48.6 | 53.1 45.5 | 52.7 50.2 |
| w/ semi-k-means | 78.1 73.8 | 81.5 73.3 | 71.3 74.8 | 54.5 48.7 | 54.1 43.9 | 54.7 51.2 |
| w/ semi-k-means$^\star$ | 76.8 71.9 | 76.9 71.8 | 76.4 72.1 | 56.6 48.1 | 57.1 50.2 | **56.4** 47.1 |
| w/ SNC (ours) | **81.5** 76.5 | 82.4 75.1 | 79.7 79.3 | **57.1** 50.2 | **58.7** 48.8 | 55.6 51.0 |

Table 6: **Results using different relations.** $u$-$u$ denotes pairwise relations from SNC between unlabelled and unlabelled data, and $u$-$\ell$ denotes pairwise relations from SNC between unlabelled and labelled data. The results evaluated with SNC are reported of normal size (left), and those with semi-supervised $k$-means are reported of smaller size (right). Rows (1)-(2) mean applying SNC on *all* data but only using $u$-$u$ or $u$-$\ell$ for pseudo positive relations. Row (3) denotes our full method.

| | $u$-$u$ | $u$-$\ell$ | CIFAR-100 | | | CUB-200 | | |
|---|---|---|---|---|---|---|---|---|
| | | | All | Seen | Unseen | All | Seen | Unseen |
| (0) | ✗ | ✗ | 73.6 70.8 | 80.4 77.6 | 60.0 57.0 | 53.1 51.3 | 57.6 56.6 | 50.8 48.7 |
| (1) | ✓ | ✗ | 80.5 76.5 | 80.6 76.3 | **80.3** 76.9 | 56.6 52.7 | 57.2 51.5 | **56.3** 53.3 |
| (2) | ✗ | ✓ | 72.9 70.0 | 82.0 79.1 | 54.8 51.6 | 51.0 45.5 | 52.9 44.4 | 50.1 46.0 |
| (3) | ✓ | ✓ | **81.5** 76.5 | **82.4** 75.1 | 79.7 79.3 | **57.1** 50.2 | **58.7** 48.8 | 55.6 51.0 |

its global centroid, SNC first connects the unlabeled instance to the close labeled instance at early clustering steps (low clustering levels) and then gradually merges it into the global labeled cluster at the final level. As a result, SNC consistently demonstrates higher accuracy compared to semi-supervised $k$-means.

**Effectiveness of cross-instance positive relations.** In this paper, we use SNC to generate pairwise relations of unlabelled data, as well as relations between unlabelled and labelled data in supervised contrastive learning. In Tab. 6, we evaluate different configurations of the pairwise relations, showing that our pairwise relation method achieves the best overall performance. Row (0) represents the performance of the state-of-the-art GCD method of Vaze et al. (2022b) without using any pseudo relations. Comparing row (1) to row (0), we can see that SNC can effectively enhance the baseline performance by solely providing pairwise relations between unlabelled data. Comparing row (2) to row (0), we can see that only adding pairwise relations of labelled and unlabelled data ($u$-$\ell$) is not sufficient to boost baseline performance and even harmful due to the biased supervision from seen categories. Row (3) is our full method that achieves the best performance, demonstrating the effective utilization of pairwise relations in our method.

## 5 Conclusion

We have presented a framework CiPR for the challenging problem of GCD. Our framework leverages the cross-instance positive relations that are obtained with SNC, an efficient parameter-free hierarchical clustering algorithm we develop for the GCD setting. Although off-the-shelf clustering methods can be utilized for generating pseudo labels, none of the current methods fulfill all of the fundamental properties of SNC at the same time, critical to GCD. The required properties consist of: (1) using label supervision for clustering unlabeled data, (2) avoiding the need for the number of clusters, and (3) obtaining the highest level of over-clustering purity, which results in reliable pairwise pseudo labels for unbiased representation learning. With the positive relations obtained by SNC, we can learn better representation for GCD, and the label assignment on the unlabelled data can be obtained from a single run of SNC, which is far more efficient than the semi-supervised $k$-means used in the state-of-the-art method. We also show that SNC can be used to estimate the unknown class number in the unlabelled data with higher efficiency.

**Acknowledgments** This work is partially supported by Hong Kong Research Grant Council - Early Career Scheme (Grant No. 27208022), National Natural Science Foundation of China (Grant No. 62306251), and HKU Seed Fund for Basic Research.

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

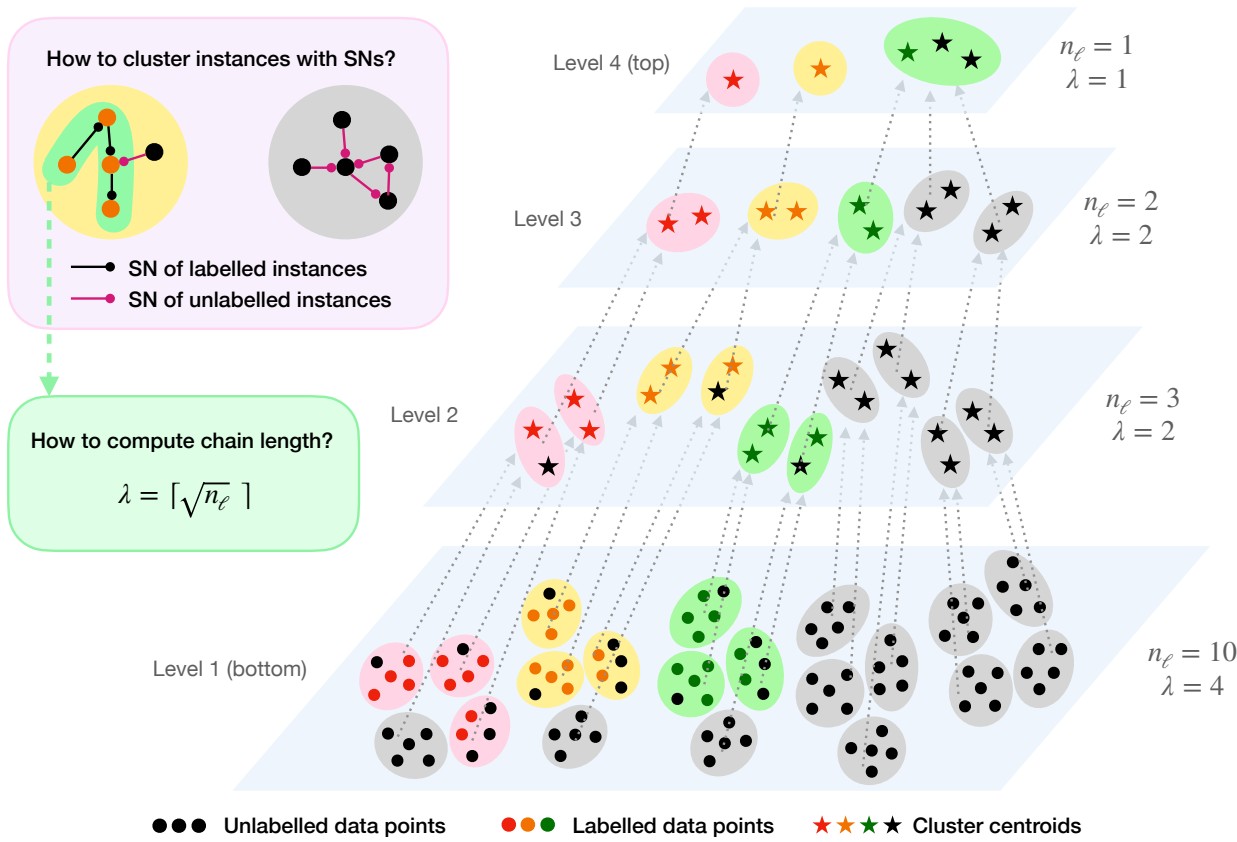

Figure 6: **A more detailed illustration of SNC.** SNC iteratively clusters instances from the bottom to the top, producing multiple levels of different partitions. At each level, the auto-adaptive chain length $\lambda$ is dynamically determined by the number of labelled 'instances' $n_\ell$ in each class. The connected components are extracted with selective neighbors (SNs), forming the clusters at each level.

## A    More analysis on SNC

We present a more detailed illustration of our proposed SNC in Fig. 6. SNC is inspired by the idea from FINCH, but they are significantly different in two key aspects: (1) FINCH treats all instances the same and simply uses nearest neighbors to construct graphs; SNC uses a novel selective neighbor strategy tailored for the GCD setting to construct graphs, treating labelled and unlabelled instances differently. (2) SNC is able to cluster a mixed set of labelled and unlabelled data by fully exploiting label supervision, but FINCH is not.

**Different choices of chain lengths $\lambda$.**    In Tab. 7, we experiment on other formulations which satisfies the above relationship, *e.g.*, $\lambda = \lceil \sqrt[3]{n_\ell} \rceil$ and $\lambda = \lceil n_\ell/2 \rceil$, and our formulation performs the best. We also compare our dynamic $\lambda$ with a possible alternative of a fixed $\lambda$. For the fixed chain length, we conduct multiple experiments with different length values to find the best length giving the highest accuracy for each dataset. We observe that the best chain length varies from dataset to dataset, and there is no single fixed $\lambda$ that gives the best performance for all datasets. In contrast, our dynamic $\lambda$ consistently outperforms the fixed one, and it can automatically adjust the chain length for different datasets and different levels, without requiring any tuning or validation like the fixed one.

**Impacts of different levels for positive relation generation.**    When the chain length is determined, clustering on different datasets will follow a similar rate, which guarantees that a fixed clustering level provides an equal clustering degree in the hierarchy for all datasets. We consider the following two conditions for the choice of a proper clustering level: (1) the number of clusters should be notably larger than the labelled class number to overcluster the instances, and (2) the number of clusters should not be too larger than the labelled class number and the batch size, which may decrease useful pairwise correlations. A proper level for positive

Table 7: **Comparison of different formulations of chain length $\lambda$.** The best fixed length values are 8 for CIFAR-100 and 3 for CUB-200.

| | CIFAR-100 | | | CUB-200 | | |
|---|---|---|---|---|---|---|
| Classes | All | Seen | Unseen | All | Seen | Unseen |
| Fixed | 80.2 | 79.7 | **81.3** | 54.3 | **58.8** | 52.1 |
| $\lceil n_\ell/2 \rceil$ | 81.4 | **84.5** | 75.2 | 45.5 | 45.5 | 45.5 |
| $\lceil \sqrt[3]{n_\ell} \rceil$ | 72.5 | 77.1 | 63.2 | 42.4 | 45.0 | 41.1 |
| $\lceil \sqrt{n_\ell} \rceil$ (ours) | **81.5** | 82.4 | 79.7 | **57.1** | 58.7 | **55.6** |

Table 8: **Comparison of different levels.** Compare levels 2, 3, 4, and baseline (Vaze et al., 2022b).

| | CIFAR-100 | | | CUB-200 | | |
|---|---|---|---|---|---|---|
| Classes | All | Seen | Unseen | All | Seen | Unseen |
| Baseline (Vaze et al., 2022b) | 70.8 | 77.6 | 57.0 | 51.3 | 56.6 | 48.7 |
| CiPR w/ level 2 | 72.4 | 79.6 | 58.0 | 50.9 | 55.8 | 48.5 |
| CiPR w/ level 3 (ours) | 81.5 | **82.4** | 79.7 | **57.1** | **58.7** | **55.6** |
| CiPR w/ level 4 | **81.6** | 81.9 | **80.8** | 52.9 | 53.1 | 52.8 |

relation generation should overcluster the labelled data to some extent, such that reliable positive relations can be generated. Level 1 is not a valid choice because no positive relations can be generated if each instance is treated as a cluster. In Tab. 8, we present the performance using levels 2, 3, and 4 to generate pseudo labels and also compare with the previous state-of-the-art baseline by Vaze et al. (2022b). We empirically find that the overclustering levels 3 and 4 are similarly good, while level 2 is worse because less positive relations are explored in each mini-batch. Even using level 2, our method still performs on par with Vaze et al. (2022b).

**Effectivenss of SNC on different learned features.** In Tab. 9, we evaluate SNC[1] on features extracted from DINO (Caron et al., 2021), GCD method of Vaze et al. (2022b), and our method CiPR. We also compare SNC with semi-supervised $k$-means (Han et al., 2019; Vaze et al., 2022b). We can observe that SNC surpasses semi-supervised $k$-means with a significant margin on all features, except those extracted by Vaze et al. (2022b) on ImageNet-100. Moreover, semi-supervised $k$-means with our features performs better than with other features. Overall, SNC with our learned features gives the best performance.

Table 9: **Effectivenss of SNC on different learned features.**

| Clustering | Features | CIFAR-100 | | | ImageNet-100 | | | CUB-200 | | | SCars | | |
|---|---|---|---|---|---|---|---|---|---|---|---|---|---|
| | | All | Seen | Unseen | All | Seen | Unseen | All | Seen | Unseen | All | Seen | Unseen |
| Semi-$k$-means | DINO (Caron et al., 2021) | 60.4 | 63.1 | 54.9 | 72.8 | 70.6 | 73.8 | 36.7 | 37.9 | 36.0 | 12.3 | 13.7 | 11.6 |
| | Vaze et al. (2022b) | 74.5 | 81.9 | 60.0 | 69.2 | 66.6 | 70.5 | 53.5 | 59.9 | 50.3 | 40.8 | **67.6** | 27.8 |
| | CiPR (ours) | 76.5 | 75.1 | 79.3 | 72.8 | 70.6 | 73.8 | 49.8 | 46.1 | 51.7 | 42.6 | 55.2 | 36.5 |
| SNC (ours) | DINO (Caron et al., 2021) | 65.5 | 69.0 | 58.3 | 76.8 | 81.1 | 74.6 | 36.7 | 35.0 | 37.5 | 12.4 | 15.8 | 10.7 |
| | Vaze et al. (2022b) | 77.8 | **87.4** | 58.6 | 61.4 | 76.7 | 53.8 | 55.9 | **61.6** | 53.0 | 41.3 | 62.9 | 30.8 |
| | CiPR (ours) | **81.5** | 82.4 | **79.7** | **80.5** | **84.9** | **78.3** | **57.1** | 58.7 | **55.6** | **47.0** | 61.5 | **40.1** |

# B A unified loss

In this paper, to leverage pseudo labels produced by SNC, we jointly train our model with two supervised contrastive losses, one using true positive relations of labelled data and the other using pseudo positive relations of all data. Indeed, it is possible to train the model with a unified loss by replacing the pseudo relations in the second term of our loss, and remove the first term. Formally, let $\mathcal{R}_\mathcal{B}(i)$ be the set of positive relations for instance $i$. The unified loss $\mathcal{L}_i^r$ can be written as

$$\mathcal{L}_i^r = -\frac{1}{|\mathcal{R}_\mathcal{B}(i)|} \sum_{q \in \mathcal{R}_\mathcal{B}(i)} \log \frac{\exp(\boldsymbol{z}_i \cdot \boldsymbol{z}_q/\tau)}{\sum_{n \in \mathcal{B}, n \neq i} \exp(\boldsymbol{z}_i \cdot \boldsymbol{z}_n/\tau)}, \tag{7}$$

---

[1]When representing a clustering method here, SNC denotes selective neighbor clustering with one-to-one merging.

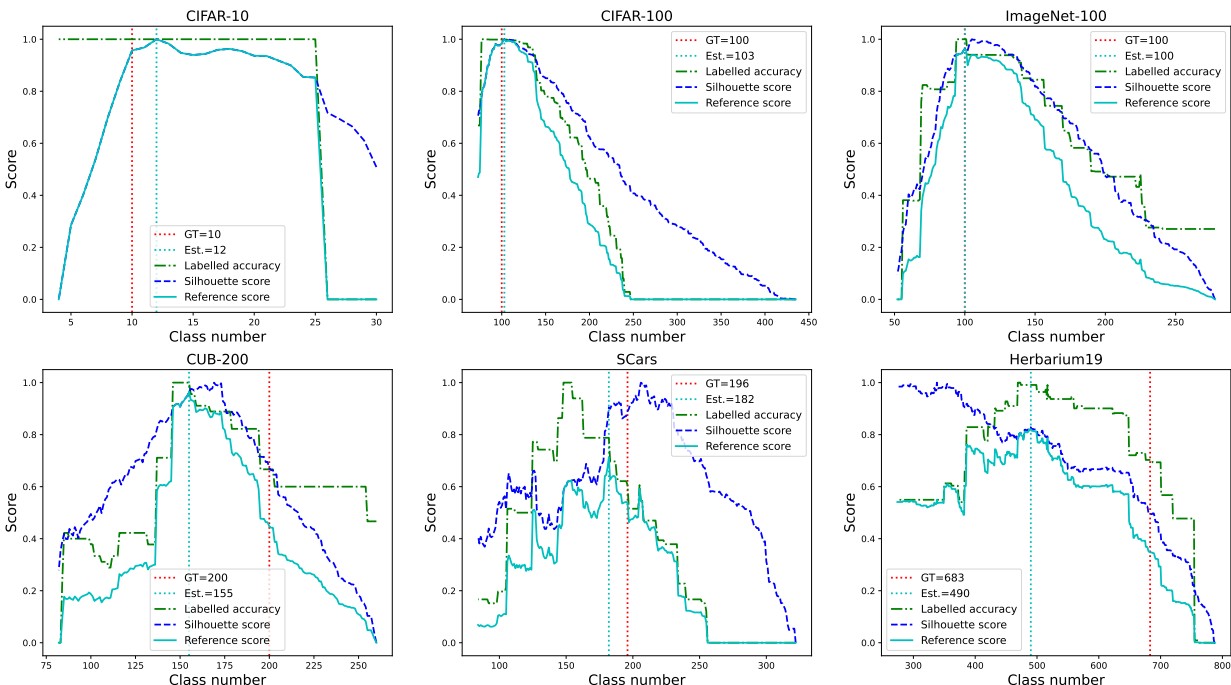

Figure 7: **Curves throughout class number estimation.** We report curves of accuracy on the labelled subset $\mathcal{D}_{\mathcal{L}}^{v}$, silhouette score on the unlabelled data $\mathcal{D}_{\mathcal{U}}$, and our reference score on $\mathcal{D}_{\mathcal{L}}^{v} \cup \mathcal{D}_{\mathcal{U}}$. The cyan vertical line denotes the estimated class number and the red vertical line denotes the ground-truth class number. Note that the $x$-axis should be read from right to left, as the merging starts from the lower level to the upper level.

where

$$
\mathcal{R}_{\mathcal{B}}(i) = \begin{cases} \mathcal{G}_{\mathcal{B}}(i) \cup (\mathcal{P}_{\mathcal{B}}(i) \cap \mathcal{I}_{\mathcal{U}}) & \text{if } i \in \mathcal{I}_{\mathcal{L}}, \\ \mathcal{P}_{\mathcal{B}}(i) & \text{if } i \in \mathcal{I}_{\mathcal{U}} \end{cases}, \tag{8}
$$

$\mathcal{I}_{\mathcal{L}}$ and $\mathcal{I}_{\mathcal{U}}$ denote the instance indices of the labelled and unlabelled set respectively. In Tab. 10, we compare our two-term loss formulation with this unified loss formulation. It turns out that our two-term loss appears to be more effective. We hypothesize the performance degradation of Eq. (7) is caused by unbalanced granularity of labelled data and unlabelled data, due to mixture of overclustering pseudo labels and non-overclustering ground-truth labels.

Table 10: **Results using different loss formulations.**.

|  | CIFAR-100 | | | CUB-200 | | |
|---|---|---|---|---|---|---|
| Classes | All | Seen | Unseen | All | Seen | Unseen |
| Eq. (7) | 79.3 | 80.3 | 77.3 | 53.9 | 53.5 | 54.0 |
| Ours (CiPR) | **81.5** | **82.4** | **79.7** | **57.1** | **58.7** | **55.6** |

## C   Class number estimation

In Fig. 7, we show how labelled accuracy, silhouette score and reference score change throughout the whole procedure of class number estimation with one-to-one merging. The accuracy on the labelled instances or the silhouette score alone does not well fit the actual cluster number. By jointly considering both, we can see the actual class number aligns well with our suggested reference score.

# D   Time efficiency

Here, we evaluate the time efficiency of CiPR, including both category discovery and class number estimation.

**Category discovery efficiency.**  The latency for the category discovery process mainly consists of two parts: feature extraction and label assignment. In Tab. 11, we present the feature extraction time. All methods consume roughly the same amount of time for feature extraction per image. RankStats+ (Han et al., 2021), UNO+ (Fini et al., 2021), and ORCA (Cao et al., 2022) assign labels with a linear classifier, thanks to the assumption of a known category number. Hence, the label assignment process is simply done by a fast feed-forward pass of a linear classifier, costing negligible time ($< 0.0005$ second per image), though their performance lags. Our CiPR and Vaze et al. (2022b) contain the transfer clustering process for label assignment, for which CiPR is 6-30 times faster than semi-supervised $k$-means used in Vaze et al. (2022b) (see Tab. 12). The high efficiency of SNC stems from the design of *hierarchical* clustering, which eliminates the need for extensive iterative steps required by methods like semi-supervised $k$-means.

Table 11: **Time cost in feature extraction per image.**

|  | Time cost |
| --- | --- |
| RankStats+ (Han et al., 2021) | 0.015s$\pm$0.001 |
| UNO+ (Fini et al., 2021) | 0.017s$\pm$0.001 |
| ORCA (Cao et al., 2022) | 0.015s$\pm$0.001 |
| Vaze et al. (2022b) | 0.014s$\pm$0.001 |
| Ours (CiPR) | 0.014s$\pm$0.001 |

Table 12: **Time cost in clustering.**

|  | CIFAR-10 | CIFAR-100 | ImageNet-100 | CUB-200 | SCars | Herbarium19 |
| --- | --- | --- | --- | --- | --- | --- |
| Semi-$k$-means | 346s | 688s | 3863s | 256s | 356s | 6053s |
| Ours (SNC w/ one-to-one merging) | 58s | 111s | 118s | 36s | 50s | 917s |

**Estimating class number.**  Compared to repeatedly running $k$-means with different class numbers as in Vaze et al. (2022b), CiPR only requires a single run to obtain the estimated class number, thus significantly increasing efficiency. In Tab. 13, CiPR is 40-150 times faster than Vaze et al. (2022b), which utilizes $k$-means with the optimization of Brent's algorithm (Brent, 1971). We also compare the memory cost. We can observe that our method costs comparable memory but achieves a much faster running speed than Vaze et al. (2022b) in class number estimation, thanks to (1) the higher efficiency of SNC and (2) the single run for our estimation method instead of multiple runs from a predefined list of possible class numbers required in Vaze et al. (2022b).

Table 13: **Time and memory consumed in estimating class number.**

|  | Method | CIFAR-10 | CIFAR-100 | ImageNet-100 | CUB-200 | SCars | Herbarium19 |
| --- | --- | --- | --- | --- | --- | --- | --- |
| Runtime (s) | Vaze et al. (2022b) | 15394 | 27755 | 64524 | 7197 | 8863 | 63901 |
|  | Ours (CiPR) | 102 | 528 | 444 | 126 | 168 | 1654 |
| Memory (MB) | Vaze et al. (2022b) | 2206 | 2207 | 3760 | 1354 | 1394 | 1902 |
|  | Ours (CiPR) | 2535 | 2932 | 5848 | 1392 | 1451 | 2205 |

# E   Comparison to concurrent SimGCD with backbone enhancement

SimGCD (Wen et al., 2023) is a concurrent work tackling GCD problem that shows competitive performance, which introduces a parametric classifier and an entropy regularization term. We further provide comparison with SimGCD using both DINOv1 (Caron et al., 2021) and recently released DINOv2 (Oquab et al., 2023) feature backbone. Our model demonstrates strong competence with both DINOv1 and DINOv2 backbones.

Table 14: **Results on generic image recognition datasets.**

| Backbone | Methods | CIFAR-10 | | | CIFAR-100 | | | ImageNet-100 | | |
|---|---|---|---|---|---|---|---|---|---|---|
| | | All | Seen | Unseen | All | Seen | Unseen | All | Seen | Unseen |
| DINOv1 | Vaze et al. (2022b) | 91.5 | 97.9 | 88.2 | 70.8 | 77.6 | 57.0 | 74.1 | 89.8 | 66.3 |
| | SimGCD (Wen et al., 2023) | 97.1 | 95.1 | 98.1 | 80.1 | 81.2 | 77.8 | 83.0 | 93.1 | 77.9 |
| | Ours (CiPR) | 97.7 | 97.5 | 97.7 | 81.5 | 82.4 | 79.7 | 80.5 | 84.9 | 78.3 |
| DINOv2 | Vaze et al. (2022b) | 97.8 | **99.0** | 97.1 | 79.6 | 84.5 | 69.9 | 78.5 | 89.5 | 73.0 |
| | SimGCD (Wen et al., 2023) | 98.8 | 96.9 | **99.7** | 88.5 | **89.3** | 86.9 | **90.4** | **96.3** | 87.4 |
| | Ours (CiPR) | **99.0** | 98.7 | 99.2 | **90.3** | 89.0 | **93.1** | 88.2 | 87.6 | **88.5** |

Table 15: **Results on fine-grained image recognition datasets.**

| Backbone | Methods | CUB-200 | | | SCars | | | Herbarium19 | | |
|---|---|---|---|---|---|---|---|---|---|---|
| | | All | Seen | Unseen | All | Seen | Unseen | All | Seen | Unseen |
| DINOv1 | Vaze et al. (2022b) | 51.3 | 56.6 | 48.7 | 39.0 | 57.6 | 29.9 | 35.4 | 51.0 | 27.0 |
| | SimGCD (Wen et al., 2023) | 60.3 | 65.6 | 57.7 | 53.8 | 71.9 | 45.0 | 44.0 | 58.0 | 36.4 |
| | Ours (CiPR) | 57.1 | 58.7 | 55.6 | 47.0 | 61.5 | 40.1 | 36.8 | 45.4 | 32.6 |
| DINOv2 | Vaze et al. (2022b) | 70.2 | 70.1 | 70.2 | 62.8 | 65.7 | 61.4 | 38.3 | 40.1 | 37.4 |
| | SimGCD (Wen et al., 2023) | 76.3 | **80.0** | 74.4 | **71.3** | **81.6** | **66.4** | 58.7 | 63.8 | 56.2 |
| | Ours (CiPR) | **78.3** | 73.4 | **80.8** | 66.7 | 77.0 | 61.8 | **59.2** | **65.0** | **56.3** |

Unsurprisingly, with the stronger DINOv2 backbone, the results are improved for all methods, while CiPR still achieves the best performance on most datasets.

## F  Data splits

In Tab. 16, we show the details on data splits of CIFAR-10 (Krizhevsky et al., 2009), CIFAR-100 (Krizhevsky et al., 2009), ImageNet-100 (Deng et al., 2009), CUB-200 (Wah et al., 2011), Stanford Cars (Krause et al., 2013) and Herbarium19 (Tan et al., 2019) in our experiments.

Table 16: **Data splits of all datasets.** We present the number of classes in the labelled and unlabelled set $(|\mathcal{Y}_\mathcal{L}|, |\mathcal{Y}_\mathcal{U}|)$, and the number of images $(|\mathcal{D}_\mathcal{L}|, |\mathcal{D}_\mathcal{U}|)$.

| | CIFAR-10 | CIFAR-100 | ImageNet-100 | CUB-200 | SCars | Herbarium19 |
|---|---|---|---|---|---|---|
| $|\mathcal{Y}_\mathcal{L}|$ | 5 | 80 | 50 | 100 | 98 | 341 |
| $|\mathcal{Y}_\mathcal{U}|$ | 10 | 100 | 100 | 200 | 196 | 683 |
| $|\mathcal{D}_\mathcal{L}|$ | 12.5k | 20k | 32.5k | 1.5k | 2.0k | 8.5k |
| $|\mathcal{D}_\mathcal{U}|$ | 37.5k | 30k | 97.5k | 4.5k | 6.1k | 25.7k |

## G  Special cases of unlabelled data

In the real world, we may meet scenarios where unlabelled data are all from seen or unseen classes. We investigate into such scenarios and conduct experiments to validate the effectiveness of our method. Our experiments are under two settings: (1) applying our pretrained models in the main paper to seen-only and unseen-only unlabelled data; (2) retraining the models with seen-only and unseen-only unlabelled data. In Tab. 17, we can observe that our model maintains strong performance in all cases.

## H  Attention map visualization

ViT (Dosovitskiy et al., 2020) has a multi-head attention design, with each head focusing on different contexts of the image. For the final block of ViT, the input $\mathbf{X} \in \mathbb{R}^{(HW+1)\times D}$, corresponding to a feature of $HW$

Table 17: **Performance on seen-only and unseen-only unlabelled data.** "original setting" denotes the performance of CiPR dealing with GCD; "direct testing" denotes the performance of CiPR dealing with seen-only or unseen-only unlabelled data using pretrained GCD model; "retraining" denotes the performance of retrained CiPR dealing with seen-only or unseen-only unlabelled data.

|  |  | CIFAR-10 | CIFAR-100 | ImageNet-100 | CUB-200 | SCars | Herbarium19 |
|---|---|---|---|---|---|---|---|
| Seen | original setting | 97.5 | 82.4 | 84.9 | 58.7 | 61.5 | 45.4 |
|  | direct testing | 98.5 | 84.4 | 83.3 | 79.1 | 72.0 | 55.4 |
|  | retraining | 98.9 | 87.0 | 87.3 | 81.9 | 75.2 | 66.3 |
| Unseen | original setting | 97.7 | 79.7 | 78.3 | 55.6 | 40.1 | 32.6 |
|  | direct testing | 97.6 | 82.7 | 74.3 | 56.5 | 39.3 | 37.9 |
|  | retraining | 98.4 | 78.9 | 79.3 | 60.4 | 42.5 | 41.3 |

patches and a `[CLS]` token, is fed into multi-heads, which can be expressed as

$$MultiHead(\mathbf{X}) = [head_1, head_2, \ldots, head_h]\mathbf{W}^O \tag{9}$$

where

$$head_j = softmax(\frac{\mathbf{Q}_j\mathbf{K}_j^T}{\sqrt{d_k}})\mathbf{V}_j \tag{10}$$

$$\mathbf{Q}_j = \mathbf{X}\mathbf{W}_j^Q \tag{11}$$

$$\mathbf{K}_j = \mathbf{X}\mathbf{W}_j^K \tag{12}$$

$$\mathbf{V}_j = \mathbf{X}\mathbf{W}_j^V \tag{13}$$

where $d_k$ is the dimension of queries and keys. In our model, patch size is $16 \times 16$ pixels and $HW = 14 \times 14 = 196$. The number of heads $h$ is 12. Referring to Vaswani et al. (2017), consider attention map of head $j$ $\mathbf{A}_j = softmax(\frac{\mathbf{Q}_j\mathbf{K}_j^T}{\sqrt{d_k}}) \in [0, 1]^{(HW+1) \times (HW+1)}$. $\mathbf{A}_j$ describes the similarity of one feature to every other feature captured in head $j$. The first row of $\mathbf{A}_j$ shows how head $j$ attends `[CLS]` token to every spatial patch of the input image. In Fig. 8, we visualize some of the interpretable attention heads to show semantic regions that ViT attends to. We can observe that our model CiPR, as well as DINO (Caron et al., 2021) and Vaze et al. (2022b), can attend to specific semantic object regions. For instance, CiPR attends three heads respectively to 'license plate', 'light' and 'wheels' for Stanford Cars (head 1 fails in row 1), and to 'body', 'head' and 'neck' for CUB-200.

## I    License for experimental datasets

All datasets used in this paper are permitted for research use. CIFAR-10 and CIFAR-100 (Krizhevsky et al., 2009) are released under MIT License, allowing for research propose. ImageNet-100 is the subset of ImageNet (Deng et al., 2009), which allows non-commercial research use. Similarly, CUB-200 (Wah et al., 2011), Stanford Cars (Krause et al., 2013) and Herbarium19 (Tan et al., 2019) are also exclusive for non-commercial research purpose.

## J    Limitation

In our current experiments, we consider images from the same curated dataset. However, in practice, we might want to transfer concepts from one dataset to another, which may have different data distributions (*e.g.*, the unlabelled data could follow the long-tailed distribution), introducing more challenges. Another limitation is that currently, we need to train the model on both labelled and unlabelled data jointly. However, in the real world, there are often cases in which we do not have access to any labelled data from the seen classes when facing unlabelled data. We consider these as our future research directions.

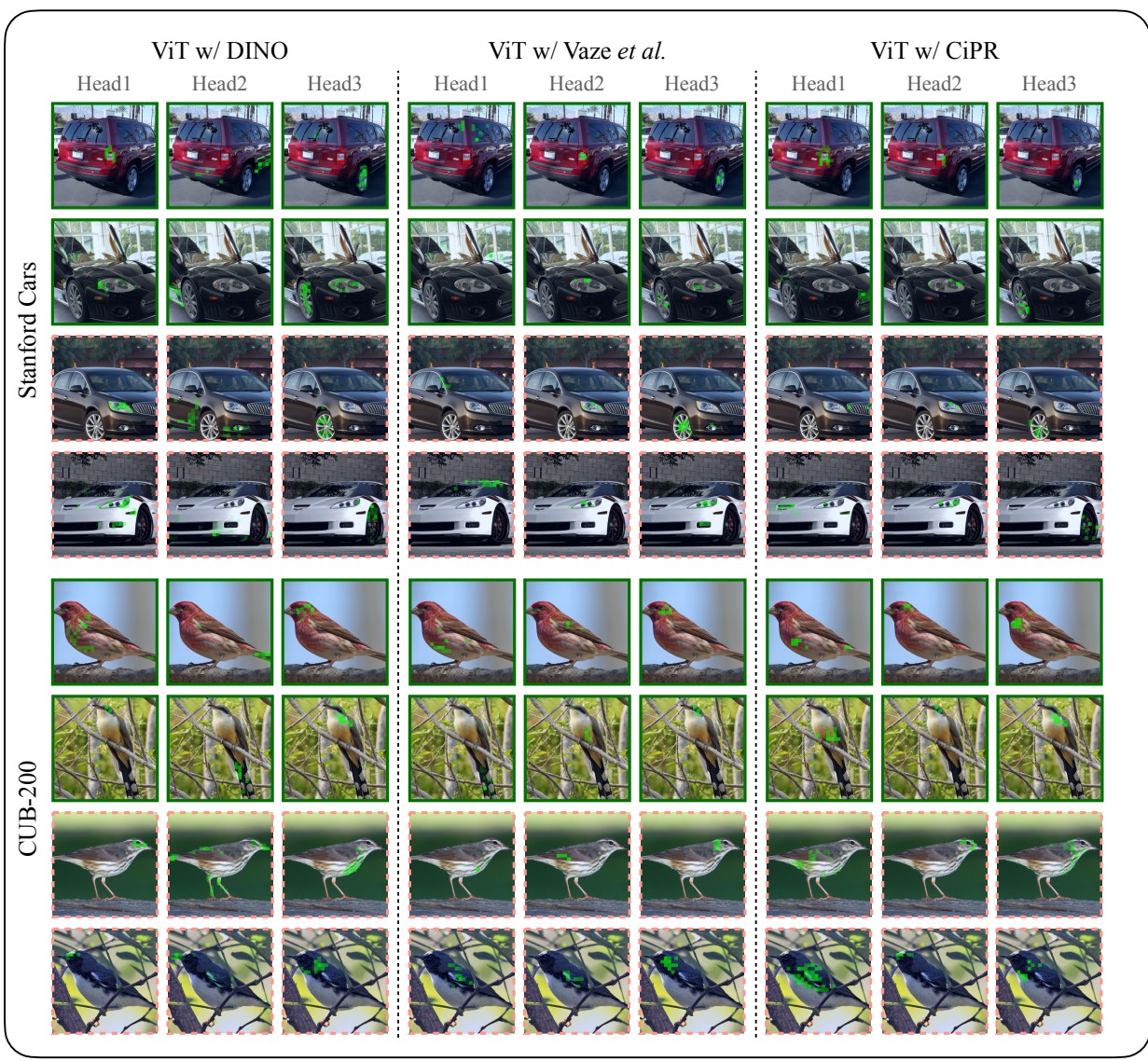

Figure 8: **Attention visualizations.** We report visualization results of DINO (Caron et al., 2021) (left), Vaze et al. (2022b) (middle), and CiPR (right) on Stanford Cars (top) and CUB-200 (bottom). For each dataset, we show two rows of 'Seen' categories (solid green box) and two rows of 'Unseen' categories (dashed red box). Zoom in to see attention details.

