# OpenReview forum: "CiPR: An Efficient Framework with Cross-instance Positive Relations for Generalized Category Discovery"
_TMLR — Accepted by TMLR_

### Review · Reviewer_ygfQ · 2023-10-08

**Summary Of Contributions:**

This paper proposes a framework of joint contrastive representation learning with cross-instance positive relations for generalized category discovery (GCD). The key of joint contrastive representation learning is to select reliable and informative pseudo labels. For that, this paper proposes selective neighbor clustering (SNC), a method for hierachical clustering of both labelled and unlabeled data. The proposed SNC method also serves to estimate the class number and assign pretictions to unlabeled data. The experiments are conducted in many GCD benchmarks and show very competitive results to previous methods.

**Audience:**

Yes

**Broader Impact Concerns:**

No.

**Claims And Evidence:**

Yes

**Requested Changes:**

See weaknesses listed above.

**Strengths And Weaknesses:**

## Strengths
- This paper proposes a framework of joint contractive representation learning by utiling both labeled and unlabeled data with cross-instance relations. This framework for GCD problem is new. We can seen from the experiments that when pseudo labels are selected well, this learning framework would be work to improve the results.
- This paper proposes a method of selective neighbor clustering (SNC), addressing the problems when labeled and unlabeled data are together for clustering. The solution for SNC is to provide $\lambda$ chain links for the labeled data, which extends the FINCH method to both labeled and unlabeled data efficiently.
- The proposed SNC method has shown many benefits to previous semi-supervised clustering method, including better pseudo labels generations and class number estimation, and faster runing time. These advantages from SNC make a non-trivial contribution to the community.
- The abalation studies and the experiments to GCD benchmarks are comprehensive and well designed, and the results of the propose method across many benchmarks are SOTA.
- Overall presentation of the paper is nice to read and the orgnization of the paper is good.
- The literature review of related works is comprehensive.

## Weaknesses
- The proposed framework of joint contrastive representation learning is somehow resonable, but I think that such framework may be very sensitive to the noisy labels. Is there any insights that such framework could be well robust to noisy labels? If not, such framework may be not very general to all problems, especilly for those cases that the initial accuray is not vey good.
- There are still some confusions about the overall method. For instance, during which stages is the SNC utilized? Is it applied after each epoch following the joint representation learning, or is it employed just once in the initial step?
- The label assignment by SNC is not very clear to understand how it works.
- Two figures in Fig. 4 are missing.

---

> ### Author Response · Authors · 2023-12-12
> **Response to Reviewer ygfQ (part 1/2)**
>
> **Thank you for your constructive comments! Below is our response.**
>
> > **W1**: The proposed framework of joint contrastive representation learning is somehow resonable, but I think that such framework may be very sensitive to the noisy labels. Is there any insights that such framework could be well robust to noisy labels? If not, such framework may be not very general to all problems, especilly for those cases that the initial accuray is not very good.
>
> Our framework offers several guarantees regarding the quality of the pseudo labels:
>
> - **Initialization with pretrained models provides high-quality clustering.** Our model is built upon the strong self-supervised pretrained model, DINO, for initialization. DINO generates highly discriminative representations, thereby establishing a foundation of strong clustering quality from the outset of the training process. DINO is pretrained using contrastive learning for instance-wise discrimination, which has been validated to be generalizable to various downstream tasks, especially discrimination tasks such as linear probing and clustering. Our framework benefits from the strong feature discrimination capabilities of DINO, leading to promisingly high-quality pseudo labels.
>
> - **SNC produces overclustered pseudo-labels, resulting in fewer noisy labels.** In self-supervised contrastive learning, all instances serve as negative samples to each other, and the introduction of pseudo-labels intrinsically adds positive samples for contrastive learning. Therefore, noisy negative samples do not undermine contrastive learning, and the quality of pseudo-labels is reflected by clustering purity.
> SNC is a purposely designed clustering method that clusters instances in a hierarchical way and autonomously generates overclustered pseudo-labels. Overclustering leads to higher cluster purity, which is beneficial to more reliable contrastive training with more accurate positive samples.
> Figure 5 in our submitted paper demonstrates that SNC exhibits high clustering purity from the beginning, which gradually improves throughout the training process. This advantage of SNC over $k$-means benefits our framework in pseudo-label robustness.
>
> - **Our framework provides a vicious circle of pseudo-labeling and representation learning.** The pseudo labeling produces reliable positive samples and then enhances the representation learning and thus further refines the pseudo labels. In this way, our framework allows for the self-evolution of pseudo labels, thereby guaranteeing their high quality.
>
> For the initialization for general clustering, we believe a favorable approach is to employ *self-supervised learning* in the early stages and use pseudo-labels on the optimized representation space after a few steps. In our framework, we find using pseudo labels from clustering on DINO features works well, so we do not conduct self-supervised learning ahead of pseudo-labeling to avoid introducing extra complexity.
>
> > **W2**: There are still some confusions about the overall method. For instance, during which stages is the SNC utilized? Is it applied after each epoch following the joint representation learning, or is it employed just once in the initial step?
>
> In our implementation, SNC is applied after (before) each epoch of joint representation learning. By integrating clustering (SNC) and representation learning in this manner, they mutually benefit from each other, leading to improved feature representations.
>
> We **have clarified** this in the revised paper (**Sec. 3.3 & 4.1**).

---

> ### Author Response · Authors · 2023-12-12
> **Response to Reviewer ygfQ (part 2/2)**
>
> > **W3**: The label assignment by SNC is not very clear to understand how it works.
>
> Given a class number, the full process of the label assignment is as follows:
> 1. Using our SNC to produce hierarchical partitions.
> 2. Applying our one-to-one merging method to merge the closest instances successively.
> 3. Running the Hungarian algorithm to find the optimal assignment between the set of cluster indices and ground truth labels.
>
> Specifically, we begin by executing SNC to produce hierarchical partitions and identify the partition level that contains the closest cluster count greater than the given class number. We then employ one-to-one merging to successively merge the two closest clusters at each iteration until the given class number is reached. Note that during the one-to-one merging process, clusters belonging to different labeled classes are not allowed to merge. The predicted cluster indices can be therefore retrieved from the final partition. We formalize the process of SNC in Algorithm 1 and the process of one-to-one merging in Algorithm 2. Lastly, we follow GCD [1] to use the Hungarian algorithm to find an optimal linear assignment from the predicted cluster indices to the ground truth labels, which produces our results of label assignment.
>
> We **have added** a more detailed description of label assignment using SNC in the revised paper (**Sec. 3.4**).
>
> > **W4**: Two figures in Fig. 4 are missing.
>
> Thanks for pointing this out. We did not realize this as the figures were displayed properly on our end during submission. We spot this issue when using the Chrome browser. We **have now resolved** this issue by rasterizing the figure, and it can now be displayed properly in the revised paper.
>
> [1] Vaze, Sagar, et al. "Generalized category discovery." CVPR. 2022.

---

### Review · Reviewer_sQhW · 2023-10-14

**Summary Of Contributions:**

In this paper, the authors propose an enhanced version of a baseline model published in 2022 for generalized category discovery. Specifically, they design a new semi-supervised clustering approach to replace the original semi-supervised $k$-means, so that there is no need to estimate the class number during training. They also show it is important to perform supervised contrastive learning on unlabeled data by using pseudo-labels generated from clustering process.

**Audience:**

Yes

**Claims And Evidence:**

Yes

**Requested Changes:**

1. Many works in semi-supervised learning have used a similar idea to incorporate unlabeled data into supervised contrastive learning. At least, there should be some discussions on these related works.

2. How are hyperparameters in the proposed approach determined? Please elaborate it.

3. why the proposed semi-supervised clustering approach can achieve better performance than semi-supervised $k$-means? Please give more discussions.

**Strengths And Weaknesses:**

Strengths:

+: The paper is written well and easy to follow. All the figures and tables are of high quality.

+: The proposed approach is technically sound. All necessary details of the proposed approach are provided.

+: The proposed approach achieves a new SOTA. Comprehensive experiments show the technical improvements against the baseline model are effective.

Weaknesses:

-: Using pseudo-labels to incorporate unlabeled data into supervised contrastive learning is not a new idea. Many works in semi-supervised learning have already adopted this idea. There is no discussion on these related works.

-: The way to search hyperparameters may not be convincing. It seems that hyperparameters (e.g. chain length, clustering level, ratio of $|\mathcal{D}^l_\mathcal{L}| : |\mathcal{D}^v_\mathcal{L}|$) are directly determined according to the reported performance (transductive accuracy).

-: It is not clear why the proposed semi-supervised clustering approach can achieve better performance than semi-supervised $k$-means. There should be more in-depth discussions and analyses.

---

> ### Author Response · Authors · 2023-12-12
> **Response to Reviewer sQhW**
>
> **We are grateful for your valuable feedback! Below is our response.**
>
> > **W1**: Using pseudo-labels to incorporate unlabeled data into supervised contrastive learning is not a new idea. Many works in semi-supervised learning have already adopted this idea. There is no discussion on these related works.
>
> Thanks for the suggestion. We **have included** a detailed discussion on the use of pseudo labels in semi-supervised learning and carefully clarified the differences between our model and the other methods in related work (**Sec. 2**) in the revised paper.
>
> > **W2**: The way to search hyperparameters may not be convincing. It seems that hyperparameters (e.g. chain length, clustering level, ratio of $|\mathcal{D}_\mathcal{L}^l|$:$|\mathcal{D}\_\mathcal{L}^v|$) are directly determined according to the reported performance (transductive accuracy).
>
> We would like to clarify that the hyperparameters are *not* decided by the reported performance. Instead, it is determined by **intuition based on the clustering process** and the **commonly used ratio setting**.
>
> We further clarify our rationale behind the hyperparameter selection process as follows:
>
> - **Chain length**: The chain length is automatically decided in each clustering (merging) hierarchy, motivated by a straight idea that it should be positively correlated with the number of labelled instances $n_l$ at the current hierarchical level. The square root of  $n_l$ is therefore the natural choice to balance the number of clustered instances in each cluster and the number of newly formed clusters. This design of chain length provides a reasonable clustering rate of labelled instances compared to the unsupervised clustering of unlabelled instances, yielding a good performance for our framework validated in Table 7.
>
> - **Clustering level**: When the chain length is decided, clustering on different datasets will follow a similar rate, which guarantees that a fixed clustering level provides an equal clustering degree in the hierarchy for all datasets. The choice of clustering level requires: *(1)* the number of clusters should be notably larger than the labelled class number to overcluster the instances, and *(2)* the number of clusters should not be too larger than the labelled class number and the batch size, which may decrease useful pairwise correlations. A level that has a cluster number 2 -10 times the number of labelled classes is proper. We empirically find level 3 consistently meets the requirements for all datasets, which is decided by the cluster number instead of the accuracy. We further demonstrate the efficiency of this choice in Table 8.
>
> - **Ratio of $|\mathcal{D}_\mathcal{L}^l|$:$|\mathcal{D}\_\mathcal{L}^v|$**: This ratio for all datasets except CIFAR-10 is set to a commonly used split ratio 8:2, which maintains a significant proportion (80%) of labelled data. The reason why we set 6:4 for CIFAR-10 is that, because CIFAR-10 has 5 labelled classes, if we set the ratio to 8:2, there will be only 1 class left in $|\mathcal{D}\_\mathcal{L}^v|$, which will lose the effect as an accuracy metric. Therefore, we eventually use our current setting of the ratio.
>
> We **have further clarified** our rationale in the revised paper (**Appendix A and Sec. 4.1**).
>
> > **W3**: It is not clear why the proposed semi-supervised clustering approach can achieve better performance than semi-supervised $k$-means. There should be more in-depth discussions and analyses.
>
> The advantage of SNC over semi-supervised $k$-means stems from its **hierarchical design**. We compare them as follows:
>
> - In semi-supervised k-means, the number of labeled centroids remains *fixed* throughout all iterations and it only considers distances between instances and centroids. This approach may prevent certain correct instance grouping in some cases. For instance, if an unlabeled instance is close to a labeled instance but far from the centroid of the labeled cluster, it may not be assigned to that class. This limitation can have a negative impact on the overall quality of the clustering results.
>
> - On the other hand, SNC is a *hierarchical* method that takes into account local distribution structures at each hierarchical layer for both labeled and unlabeled data. In the scenario mentioned above, where an unlabeled instance is close to a labeled instance but distant from its global centroid, SNC first connects the unlabeled instance to the close labeled instance at early clustering steps (low clustering levels) and then gradually merges it into the global labeled cluster at the final level.
>
> As a result, our SNC consistently demonstrates higher accuracy compared to semi-supervised $k$-means in our experiments. Additionally, SNC proves to be more time-efficient than semi-supervised $k$-means in both clustering and class number estimation attributed to its hierarchical computing, as indicated in Tables 12 & 13. We **have added** the discussion and analysis in the revised paper (**Sec. 4.4**).

---

> ### Author Response · Authors · 2024-01-20
> **Updated revision**
>
> We have moved the illustration of the rationale behind the chain length from Appendix A to **Sec. 3.3** in the main paper. The new revision is marked in teal. Thank you!

---

### Review · Reviewer_sNyL · 2024-01-15

**Summary Of Contributions:**

This paper tackles the Generalized Category Discovery (GCD) problem in a partially labeled dataset, where unlabeled data may include instances from both known and novel categories. It proposes CiPR, a framework that utilizes Cross-instance Positive Relations for contrastive learning, a facet often overlooked in existing methods. To enhance representation learning, the paper introduces Selective Neighbor Clustering (SNC), a semi-supervised hierarchical clustering algorithm, that generates a clustering hierarchy directly from a graph constructed from selective neighbors. A method for estimating the unknown class number using SNC is also presented, extending it to facilitate label assignment for unlabeled instances. Evaluating on public image recognition datasets, CiPR with SNC establishes a new state-of-the-art in addressing the GCD problem with an unknown category number in partially labeled datasets.

**Audience:**

Yes

**Broader Impact Concerns:**

NA.

**Claims And Evidence:**

Yes

**Requested Changes:**

- Could the paper provide more details and intuitions about why the proposed method does not need the known class number?
- Is the research topic related to Out-of-distribution detection [R1,R2]?
- Why $\lambda$ is set to $\sqrt{n_l}$? I do not understand the intuition of the setting.
- The proposed method relies on K-means. However, from the reported results, it seems that the proposed method does not suffer from a high time complexity. Could the paper provide more details about this?

Overall, I think this is a good paper. A round of revision can further improve it.

----
[R1] Out-of-distribution detection with an adaptive likelihood ratio on informative hierarchical vae. NeurIPS 2022.
[R2] Out-of-distribution detection learning with unreliable out-of-distribution sources. NeurIPS 2023.

**Strengths And Weaknesses:**

**Strengths**
- This paper studies a realistic and important problem.
- The motivation of this paper is clear.
- The reported results are overall great. Although in some cases, the proposed method cannot work very well, it can achieve the best performance in most experimental cases.

**Weaknesses**
- The writing of this paper can be further improved before acceptance. There are a series of unclear descriptions and justifications in the current form, which need to be modified.

---

> ### Author Response · Authors · 2024-01-20
> **Response to Reviewer sNyL**
>
> **We thank the valuable suggestions on improving the writing of our paper!**
>
> Following the suggestions, we have revised the paper by adding:
> - The details about why SNC does not need a known class number (**Sec. 3.4**).
> - The description of OOD, including [R1,R2], in the related work (**Sec. 2**).
> - The intuition of the chain length $\lambda$ (**Sec. 3.3**).
> - The illustration of the high time efficiency of our method (**Appendix D**). Note that our method relies on **a well-designed hierarchical clustering method SNC instead of the centroid-based method k-means**.
>
> The new revisions are marked in teal.

---

### Author Response · Authors · 2024-01-20
**Global comment**

Dear reviewers,

**Thank you for your constructive comments!**

We have carefully addressed individual concerns in the response to each reviewer and followed the suggestions to enhance the paper. Here, we summarize the revisions:

*The first-round revisions are marked in blue*

1. We have clarified that SNC is used before each training epoch in **Sec. 3.3 & 4.1**. [`ygfQ`]
2. We have added a more detailed description of label assignment using SNC in **Sec. 3.4**. [`ygfQ`]
3. We have updated **Fig. 4**. [`ygfQ`]
4. We have carefully discussed the use of pseudo labels in semi-supervised learning and clarified the differences between our model and the other methods in **Sec. 2**. [`sQhW`]
5. We have clarified the rationale behind the clustering level in **Appendix A**. [`sQhW`]
6. We have clarified the rationale behind the ratio of $|\mathcal{D}_\mathcal{L}^l|$:$|\mathcal{D}\_\mathcal{L}^v|$ in **Sec. 4.1**. [`sQhW`]
7. We have analyzed why SNC demonstrates higher accuracy compared to semi-supervised $k$-means in **Sec. 4.4**. [`sQhW`]

*The second-round revisions are marked in teal*

8. We have clarified the rationale behind the chain length $\lambda$ in **Sec. 3.3**. [`sQhW`,`sNyL`]
9. We have discussed why SNC does not need a known class number in **Sec. 3.4**. [`sNyL`]
10. We have discussed OOD in **Sec. 2**. [`sNyL`]
11. We have illustrated the high time efficiency of our method in **Appendix D**. [`sNyL`]

***
We would like to thank you again for the valuable suggestions, which have helped us further strengthen our paper. If you still have any concerns or suggestions, we would be happy to discuss them!

---

### Decision · Action_Editor_evCG · 2024-02-24

**Recommendation:** Accept as is

**Comment:**

The authors incorporated the reviewer feedback in the manuscript already and the current version looks suitable for publication.

**Audience:**

Generalised category discovery is a research area that can attract a relatively wide audience, as it combines self-/semi- supervised learning with clustering and could potentially be applied for many tasks at the class/object/pixel leve.

**Claims And Evidence:**

The paper explores generalised category discovery in a partially labeled dataset, and offers a well performing method for an upcoming and exciting research direction. All reviewers recommend acceptance.